# Teaching VLMs to Admit Uncertainty in OCR from Lossy Visual Inputs

**Shuhao Guan**[1]    **Moule Lin**[2]    **Cheng Xu**[1]    **Jinman Zhao**[3]    **Derek Greene**[1]

[1] School of Computer Science, University College Dublin,
[2] Trinity College Dublin and Lero, [3] University of Toronto

## Abstract

Vision-language models (VLMs) are increasingly replacing traditional OCR pipelines. However, they often hallucinate on lossy visual inputs, such as visually degraded document images, producing fluent yet incorrect text without signaling uncertainty. This occurs because current post-training emphasizes accuracy, which encourages models to guess even when uncertain. The problem persists in state-of-the-art systems and severely impacts OCR reliability. To improve the trustworthiness of OCR on degraded documents, we propose uncertainty-aware OCR. Rather than suppressing guesses, our model transcribes while explicitly bracketing spans it deems unreliable with uncertainty tags. To train our model, we use Group Relative Policy Optimization (GRPO). We define usage rules for uncertainty tags and an evaluation protocol, introducing a pseudo-labeled cold start and a multi-objective reward that balances transcription accuracy and uncertainty coverage while preventing reward hacking. We explore different combinations of cold-start and reward granularity. We also assess the effect of reward parameters in preventing reward hacking and improving the corresponding metrics. Furthermore, we introduce Blur-OCR, a challenging benchmark for uncertainty-aware OCR on degraded document images under lossy visual conditions. In extensive experiments, our model maintains transcription accuracy while achieving an uncertainty tag F1 score of 0.685. Data and code are available at `https://github.com/NikoGuan/Uncertainty_OCR`.

## 1 Introduction

Transformer-based vision-language models (VLMs) are now central to modern OCR systems (Vaswani et al., 2017; Blecher et al., 2023; Wei et al., 2024; Feng et al., 2025). However, these models frequently hallucinate when faced with lossy visual inputs, such as degraded historical documents, generating plausible text that is unsupported by the visual evidence and does not indicate uncertainty (Kalai et al., 2025; Kaltchenko, 2025), including in unreadable regions. Such inputs arise from blur, low resolution, compression artifacts, occlusion, or severe degradation, where visual evidence is insufficient for faithful transcription (Guan & Greene, 2024). In contrast, classical OCR models tend to emit garbled or ill-formed strings in unreadable areas, which are easier to localize and thus less likely to be incorrectly used or widely propagated (Jatowt et al., 2019). VLM OCR errors, by comparison, often appear plausible (Yang et al., 2024), making error localization and correction difficult without external knowledge, which in turn increases the risk of error propagation and can severely degrade downstream analytics, model training, and cultural-heritage digitization (Hamdi et al., 2022; Zhang et al., 2024; Oelschlager, 2024; Kang et al., 2026). Despite these issues, VLMs substantially outperform traditional OCR models on many benchmarks (Liu et al., 2024; Wei et al., 2024; Guan et al., 2024), motivating the need for Transformer-based OCR methods that can identify and explicitly mark localized uncertainty while maintaining strong transcription accuracy.

In this paper, we present an *uncertainty-aware OCR* paradigm in which the model transcribes text while also enclosing spans it deems uncertain or potentially erroneous within paired delimiters: `<C>...</C>` uncertainty (UNC) tags. This approach does not suppress guesses but instead makes them transparent by marking self-identified uncertainty. We formalize tag semantics and member-

ship rules at both character and word granularities, together with an evaluation protocol reporting precision/recall/F1 for uncertainty tagging alongside transcription standard accuracy metrics.

To train a model with this behavior, we employ a pseudo-labeled cold start followed by Group Relative Policy Optimization (GRPO) (Shao et al., 2024), together with a multi-objective reward that balances transcription accuracy and uncertainty coverage while preventing reward hacking. We study tag granularity at both character and word levels, uncovering a counterintuitive yet effective configuration: a character-level cold start paired with a word-level reward yields the best calibration. We also introduce *Blur-OCR*, a challenging OCR benchmark of document images with highly diverse synthetic degradations, which includes 107,520 training images and 2,048 evaluation images, and is designed to test our uncertainty-aware OCR on degraded documents.

This paper is structured as follows. Section 3 defines the usage and membership rules for UNC tags. Section 4 introduces evaluation metrics for uncertainty-aware OCR. Section 5 presents our pseudo-labeled cold-start procedure and the multi-objective reward that prevents reward hacking. Section 6 describes the new Blur-OCR benchmark dataset, and Section 7 reports results from extensive experiments. We demonstrate the importance of appropriate cold-start design for final performance, verify the benefits of GRPO, and analyze the impact of reward parameters. We also evaluate multiple open- and closed-source models on Blur-OCR and compare several uncertainty-estimation approaches. Our model shows strong transcription accuracy while achieving uncertainty-tag precision of 0.839 and recall of 0.620, substantially outperforming all baselines.

## 2  RELATED WORK

Early neural OCR methods commonly followed a two-stage "detect → recognize" paradigm. For instance, CTPN (Tian et al., 2016) and EAST (Zhou et al., 2017) proposed text regions, while DB-Net (Liao et al., 2020) introduced differentiable binarization to improve recall on irregular shapes. For recognition, CRNN popularized CTC-based sequence modeling (Shi et al., 2016), and later attention/seq2seq systems such as SAR (Li et al., 2019) and ASTER (Shi et al., 2018) improved robustness to perspective and curved text via 2D attention and rectification modules. Benchmarks like ICDAR-MLT (Nayef et al., 2019) tracked rapid progress in multilingual settings. Transformer-based, end-to-end recognizers then displaced RNNs: TrOCR (Li et al., 2023) framed OCR as a ViT-encoder/decoder-LM sequence-to-sequence problem; Donut (Kim et al., 2022) took an "OCR-free" path that directly generated structured outputs from page images; and Nougat (Blecher et al., 2023) specialized to scientific documents, producing LaTeX-like markup. In parallel, document-aware large language models (LLMs) such as Pix2Struct (Lee et al., 2023) and PaLI (Chen et al., 2022) have increasingly integrated OCR capabilities, though community evaluations (OCRBench / OCRBench v2) show persistent challenges on noisy, degraded, or fine-grained text (Liu et al., 2024; Fu et al., 2024; Yu et al., 2025). "OCR-2.0" efforts pursue unified, end-to-end systems: GOT (Wei et al., 2024) unifies text, formulas, tables, and other character-like signals within a single model and supports interactive region/multipage use; Dolphin (Feng et al., 2025) and MinerU2.5 (Niu et al., 2025) follow an analyze-then-parse paradigm. Meanwhile, general LLMs, such as ChatGPT and Gemini, continue to advance on OCR (Hurst et al., 2024; Team et al., 2024; Yang et al., 2024).

Classical OCR methods exposed character/word confidences that enabled triage, human-in-the-loop review, and downstream re-weighting (Mor & Wolf, 2018). Empirical studies have shown that these scores correlate with errors and, after per-engine calibration, can prioritize low-quality pages for re-OCR (Cuper et al., 2023). In scene text recognition, some work calibrated predictions at the word/sequence level (Slossberg et al., 2022), while other work treated confidence-based error detection as text classification by feeding classical OCR outputs (with engine confidences) into a BERT detector (Hemmer et al., 2024). For large models, uncertainty signals have been investigated via entropy or output disagreement (Kaltchenko, 2025; Ajayi et al., 2025; Zhang et al., 2025). Zhang et al. (2025) use OCR output disagreement to detect and correct OCR errors. He et al. (2025) targeted Visual Question Answering from degraded documents with GRPO and refusal/self-awareness to limit hallucinations in answers. There is also increasing interest in uncertainty for VLMs (Kostumov et al., 2024; Fang et al., 2025; Yang et al., 2025; Lin et al., 2025).

In LLM post-training, RL has enabled optimization of sequence-level, non-differentiable objectives beyond token likelihood. Preference-based RL (RLHF/RLAIF) often uses PPO (Schulman et al., 2017), while preference matching without a learned critic (e.g., DPO) is an alternative option

(Rafailov et al., 2023). In our work, we adopt GRPO (Shao et al., 2024), which replaces a value model with group-relative advantages, making it convenient to optimize a composite reward that combines transcription accuracy with uncertainty-coverage terms. This fits naturally into the Reinforcement Learning with Verifiable Rewards (RLVR) paradigm (Lambert et al., 2024), since both components of the reward can be computed from the ground-truth transcripts.

## 3 UNCERTAINTY TAGGING

In this paper, a span is uncertain if the model judges that its transcription may not match the ground-truth text given the image evidence. A UNC tag is defined as the paired delimiter `<C>` ... `</C>` which the model emits to enclose spans it considers uncertain or possibly erroneous. At inference time, the model expresses this belief by bracketing such spans with paired UNC tags. To avoid ambiguity in evaluation and for the purpose of describing a coverage-based reward later in Section 5.2, we first formalize the semantics of UNC tags and the rules for what counts as inside vs. outside. We evaluate at two granularities $L \in \{\text{char}, \text{word}\}$. In the former, we operate at the character level; in the latter, at the word level with whitespace tokenization.

**Tag validity and alignment.** Since UNC tags function as brackets and models may emit invalid structures, we first drop all invalid tags (nested, overlapping, or unclosed) and keep only well-formed pairs. We then align the prediction to the ground truth (GT) at the chosen granularity to determine, for each character or word, whether it falls inside or outside UNC.

**Character level.** In model prediction $\hat{y}$, a character is explicitly inside UNC if it lies between a paired `<C>` and `</C>`. A contiguous GT segment that is present in GT $y$ but missing in $\hat{y}$ is treated as implicitly inside UNC if, on the alignment path, the gap lies directly adjacent to the boundary of an explicit UNC span (i.e., there are no aligned characters in between). All other positions are outside. Thus, the total number of characters counted as inside UNC is the sum of explicitly tagged characters and implicitly attributed missing characters.

**Word level.** A word is considered explicitly inside UNC if at least one of its characters is explicitly inside at the character level, with the span expanded to cover full word boundaries. For GT missing segments, we mirror the character-level rule at the word granularity. Thus, any word-level gap that is adjacent to an explicit UNC span (allowing intervening whitespace) is implicitly counted as inside.

## 4 METRICS

Building on the definitions in Section 3, we now present evaluation metrics aligned with the two core goals of uncertainty-aware OCR: (i) transcription accuracy, and (ii) correctness of UNC tag usage. We compute both at each granularity $L$. We report both metrics as macro averages.

**Transcription accuracy.** Before calculating transcription accuracy, we apply minimal normalization to remove newline characters, and collapse consecutive spaces into a single space. We then strip all UNC tags and compute standard error rates. Let $\text{ED}_L(y, \hat{y})$ be the Levenshtein edit distance at level $L$, and $|y|_L$ the length of the reference in units of $L$. We then define error rate and accuracy as:

$$e_L = \frac{\text{ED}_L(y, \hat{y})}{|y|_L}, \qquad \text{Accuracy}_L = 1 - e_L$$

**UNC precision, recall and $F_1$.** To assess tag usage, we treat "error" as the positive class and cast the inside/outside decision as a binary classifier over units at level $L$. Let $\text{ErrIn}_L$ be erroneous units inside UNC (true positives), $\text{CorrectIn}_L$ be correct units inside UNC (false positives), and $\text{ErrOut}_L$ be erroneous units outside UNC (false negatives). Then Precision $P_L$, recall $R_L$, and $F_{1,L}$ are:

$$P_L = \frac{\text{ErrIn}_L}{\text{CorrectIn}_L + \text{ErrIn}_L}, \qquad R_L = \frac{\text{ErrIn}_L}{\text{ErrOut}_L + \text{ErrIn}_L}, \qquad F_{1,L} = \frac{2 P_L R_L}{P_L + R_L}.$$

All counts are computed under the UNC membership rule above (explicit spans plus the implicit coverage of adjacent gaps) at the chosen granularity $L$. By construction, the total number of erroneous units equals the edit distance at level $L$, so $R_L$ measures the fraction of all errors covered by UNC, while $P_L$ measures the purity of what UNC encloses.

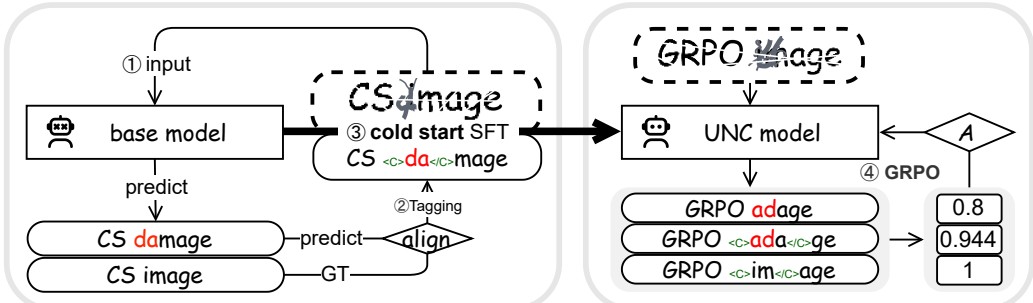

Figure 1: Uncertainty-aware OCR pipeline: Cold-start (left) and GRPO (right). ① Run the frozen base model on degraded CS images to obtain transcripts. ② Align each prediction with the ground truth (GT) and insert UNC tags (character-level shown). ③ Cold-start SFT on (image, OCR-with-tags) to obtain a tag-aware UNC model. ④ GRPO on a disjoint RL set: sample multiple completions, compute the composite reward, convert to group-relative advantages $A$, and update the model.

**UNC error rate gap.** We also report a gap metric that compares the error rate inside UNC spans to that outside. Let $|y|_{\text{in},L}$ and $|y|_{\text{out},L}$ be the GT lengths assigned to the inside and outside regions under the UNC membership rule (Sec. 3). We define

$$e_{\text{in},L} = \frac{\text{ErrIn}_L}{|y|_{\text{in},L}}, \qquad e_{\text{out},L} = \frac{\text{ErrOut}_L}{|y|_{\text{out},L}}, \qquad \text{Gap}_L = e_{\text{in},L} - e_{\text{out},L}.$$

If $|y|_{\text{in},L} = 0$ or $|y|_{\text{out},L} = 0$, we report the corresponding rate and $\text{Gap}_L$ as n/a. Since insertions increase $\text{ErrIn}_L$ or $\text{ErrOut}_L$ without increasing the corresponding GT length, $e_{\text{in},L}$, $e_{\text{out},L}$, and thus $\text{Gap}_L$ can exceed 1.

## 5 METHODS

We frame uncertainty-aware OCR as a sequence decision task. As defined in Section 3, UNC tags are paired brackets that can span long text, so their placement is a global choice that depends on the entire transcript. Token-level SFT is therefore misaligned: it optimizes local next-token likelihood, whereas our evaluation targets span-level uncertainty quality together with transcription accuracy. Local likelihood can over-penalize globally better transcripts (with near-correct spans) and under-penalize globally worse ones (e.g., invalid or truncated tags). This section presents a two-phase training pipeline (Fig. 1), cold start (CS) followed by GRPO-which optimizes a multi-objective reward balancing transcription accuracy and uncertainty coverage while mitigating reward hacking.

### 5.1 PSEUDO-LABELED COLD START

Reinforcement learning alone is insufficient to teach a VLM: without prior exposure, the policy rarely explores valid tag structures, convergence slows, and calibration is poor (Stiennon et al., 2020; Ouyang et al., 2022). We therefore initialize training with a novel pseudo-labeled cold start.

The cold-start supervision we seek consists of degraded images paired with transcripts that may contain errors, with these errors wrapped in UNC tags. Our goal is not to train the model to tag all text arising from visually degraded regions, but rather to tag spans where its own predictions are unreliable. Since VLMs can rely on both visual evidence and textual context, a model may confidently infer the correct content from context even in visually unreadable regions. In such cases, we do not want the model to add a UNC tag, because the resulting transcription is still correct and useful to downstream users. Concretely, we run a pretrained base model on a set of degraded images, align each prediction to the ground-truth transcript to localize substitutions, insertions, and deletions, and then insert UNC tags into the model's own transcript around the detected errors.

We prepare labels at two granularities: (i) character level, where tags precisely enclose erroneous characters from substitutions or insertions, and (ii) word level, where any word containing at least one error is entirely wrapped in tags. For both granularities, missing characters or words in the prediction have no corresponding text in the transcript and therefore cannot be tagged. To address

this, we assign the nearest word to carry the uncertainty and wrap that word with UNC tags. We then perform SFT on a base model using pairs of (image, OCR-with-tags) for CS. Examples of CS labels are provided in Appendix C. Starting GRPO from this tag-aware initialization provides informative signals for both accuracy and tagging, which stabilizes training and accelerates convergence.

We tag the model's output, not the GT. At inference the model only has its own transcript, not the GT; if we tagged the GT, the model would learn to place tags at GT positions it never sees at test time, and insertions/deletions would break the mapping. Tagging its own output keeps training and inference consistent. Note that SFT on model outputs carries a risk of reducing transcription accuracy, so during SFT we also mix in a subset of (image, GT) pairs to prevent a large drop in accuracy; accuracy can be recovered with following GRPO, we discuss this effect in Section 7.1.

## 5.2 UNCERTAINTY-AWARE OCR REWARD

We seek to reinforce two capabilities established during the cold-start process: (i) accurate transcription and (ii) tagging uncertain or potentially erroneous spans with UNC. We do not penalize tag-format mistakes in the reward, because the policy may emit tags multiple times and malformed tags are common; penalizing them would discourage tag usage. Instead, we ignore invalid tags during evaluation and reward computation, as we discussed in Section 3. Therefore, we combine accuracy and tagging quality into a single reward function:

$$\mathcal{R}_L(\hat{y}, y) \; = \; \max\Big\{ 0, \; (1 - e_L) \; + \; \lambda\,\eta\,e_L\,F_{\beta,L} \Big\}.$$

**Transcription term.** This is simply $(1-e_L)$. Since $e_L$ can exceed 1 on difficult examples, $(1-e_L)$ may become negative; the outer $\max\{0, \cdot\}$ clamps the total reward to be nonnegative for stability.

**Tagging term.** The tagging score is $F_{\beta,L}$, the parameter $\beta$ controls the trade-off: $\beta < 1$ biases precision, $\beta > 1$ biases recall which is equivalent to the $\beta$-weighted generalization of the $F_1$ measure

$$F_{\beta,L} \; = \; \frac{(1 + \beta^2)\,P_L\,R_L}{\beta^2 P_L + R_L},$$

**Length-mismatch damping $\eta$.** When the hypothesis length diverges substantially from the reference, a tag placed near a boundary can, under our implicit-coverage rule, "cover" large adjacent gaps and thus inflate the tagging score despite poor transcription. To neutralize this failure mode, we downweight the tagging term only under extreme length mismatch. Specifically, we define

$$\rho_L \; = \; \frac{\max\big(|\hat{y}|_L,\, |y|_L\big)}{\min\big(|\hat{y}|_L,\, |y|_L\big)}, \qquad \eta \; = \; 2^{-\mathbb{I}[\rho_L > \tau]},$$

where $\mathbb{I}[\cdot]$ is the indicator (true = 1, false = 0). Thus $\eta = \frac{1}{2}$ only when $\rho_L > \tau$ (extreme length mismatch), and $\eta = 1$ otherwise; we set $\tau = 1.3$ in all experiments. This damping is necessary because, under extreme mismatch, a boundary UNC span can cover a large adjacent gap when $|\hat{y}|_L \ll |y|_L$ or a large surplus when $|\hat{y}|_L \gg |y|_L$. In both cases many tokens are counted as "correctly tagged" despite poor transcription, which inflates the tagging term. The factor $\eta$ halves the tagging contribution only when $\rho_L > \tau$, preventing reward hacking via truncation or over-generation. We vary $\eta$ in Exp. 7.3.

**No incentive to manufacture errors.** For notational convenience, we omit $L$ below; the statements hold at both granularities. Ignoring the outer $\max\{\cdot, 0\}$ for an upper-bound argument, since $0 \leq F_\beta \leq 1$ and $\eta \leq 1$,

$$\mathcal{R}(\hat{y}, y) \leq (1 - e) + \lambda e \; = \; 1 - (1 - \lambda)e =: \mathcal{R}_{\max}(e).$$

If $0 < \lambda < 1$, $\mathcal{R}_{\max}(e)$ strictly decreases in $e$ because $\frac{d\,\mathcal{R}_{\max}}{d\,e} = \lambda - 1 < 0$. Discretely, for one extra unit error $\Delta e = \frac{1}{|y|}$ that is perfectly flagged,

$$\Delta\mathcal{R} \leq -\Delta e + \lambda\,\Delta e = (\lambda - 1)\Delta e < 0,$$

so increasing errors always reduces the reward, even under perfect tagging. So we set $0 < \lambda < 1$.

## 5.3 GROUP RELATIVE POLICY OPTIMIZATION

Using the sequence-level alignment-based reward described above, we train the policy with Group Relative Policy Optimization (GRPO) (Shao et al., 2024). GRPO samples multiple completions for the same input, ranks them by reward, and converts the relative differences into normalized advantages, which removes the need for a learned critic and reduces sensitivity to the absolute reward scale. These properties suit our uncertainty-aware OCR setting, where rewards can be noisy and difficult to calibrate.

For each input $q_b$ with reference $y_b$, we sample a group of $G$ completions $\{\hat{y}_{b,1}, \ldots, \hat{y}_{b,G}\}$ from the behavior policy $\pi_{\text{old}}$ (from the previous iteration). Each completion receives a scalar sequence-level reward $\mathcal{R}_L(\hat{y}_{b,g}, y_b)$, we then standardize rewards within the group to form relative advantages $A_{b,g}$.

The policy $\pi_\theta$ is updated by maximizing a clipped GRPO objective with a KL regularizer to reference model $\pi_{\text{ref}}$

$$\mathcal{J}_{\text{GRPO}}(\theta) = \frac{1}{B} \sum_{b=1}^{B} \frac{1}{G} \sum_{g=1}^{G} \left[ \min\left( \rho_{b,g}\, A_{b,g},\ \text{clip}(\rho_{b,g},\, 1{-}\epsilon,\, 1{+}\epsilon)\, A_{b,g} \right) - \beta_{KL}\, \mathbb{D}_{\text{KL}}\big( \pi_\theta \,\|\, \pi_{\text{ref}} \big) \right],$$

where $B$ is the number of inputs per batch, $\epsilon$ is the clipping range for the likelihood ratio, and $\beta_{KL}$ weights the KL penalty toward the reference. We use the sequence-level likelihood ratio $\rho_{b,g} = \frac{\pi_\theta(\hat{y}_{b,g}|q_b)}{\pi_{\text{old}}(\hat{y}_{b,g}|q_b)}$. The divergence $\mathbb{D}_{\text{KL}}[\pi_\theta \| \pi_{\text{ref}}]$ is computed per token and averaged over the sequence.

## 6 BLUR-OCR BENCHMARK

To support the experiments in Section 7, we construct document images that are deliberately challenging for OCR and prone to VLM hallucination. Obtaining a large-scale, diverse set of real degraded documents with GT annotations is difficult. As an alternative, prior work has shown that adding synthetic degradations to document images effectively mimics real degradation (Blecher et al., 2023; Guan et al., 2025; Yu et al., 2025; He et al., 2025). We therefore adopt a synthetic-degradation pipeline to produce a large set of degraded document images. We collect source materials from Project Gutenberg, covering diverse genres, fonts, and publication eras, with layouts varying in line length, line spacing, and character spacing. Following the Prep-OCR corruption method (Guan et al., 2025), we apply a sequence of controlled corruptions in random order and with randomized strengths: additive noise, resolution reduction, Gaussian blur, background textures, stain overlays, and thin black/white lines that mimic scratches or folds. We also introduce random opaque or transparent patches of varying sizes and apply morphological operations to simulate broken strokes and ink bleed. Finally, after these global perturbations, we select several random regions on each page for strong local degradation. The full set of parameter settings for each degradation operation is listed in Appendix G. We create an evaluation benchmark of 2,048 images with paired GT transcripts, referred to as *Blur-OCR*. The training split contains 107,520 images but with disjoint source data. To the best of our knowledge, this is the first benchmark dataset for uncertainty-aware OCR transcription on degraded documents. Figure 2 reports the per-page text-length distribution and shows sample pages.

In the Blur-OCR benchmark, we do not explicitly annotate which regions are "truly unreadable." The notion of "unreadable" is inherently model- and observer-dependent: different VLMs or human readers may disagree on which regions are still decipherable. Some datasets (e.g., KIE-HVQA) use explicit region-level labels tied to synthetic occlusions that equate unreadability with specific visual artifacts (He et al., 2025); we instead define uncertainty at the transcript level to prevent overfitting to specific visual patterns. This approach focuses on the correctness and uncertainty of the output text, rather than on pixel-level readability of the image.

## 7 EXPERIMENTS

### 7.1 EXP1: SFT VS. GRPO AND THE EFFECT OF UNCERTAINTY TAGS ON OCR ACCURACY

**Setup.** In this experiment, we want to study (i) whether GRPO improves OCR accuracy beyond SFT and (ii) whether learning to emit UNC tags affects accuracy while enabling reliable tagging.

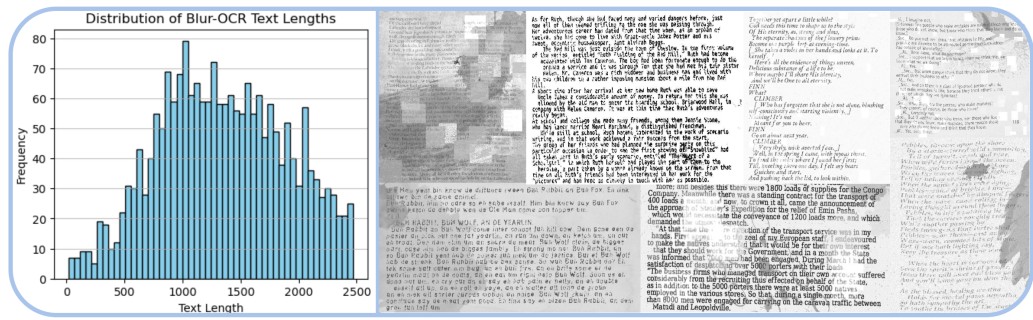

Figure 2: Per-page text-length distribution and example pages from the Blur-OCR benchmark.

We train four variants of the same backbone (Qwen2.5-VL-7B) and evaluate on Blur-OCR. Unless noted otherwise, cold-start auto-tagging and the GRPO reward operate at the character level; GRPO uses $\lambda=0.9$ and $\beta=1.0$.

**Model variants.** M1 (w/o UNC, SFT): supervised fine-tuning on 48k image–GT pairs without tags (plain transcripts), producing an OCR-strong baseline. M2 (w/o UNC, SFT→GRPO): continue from M1 and run GRPO on 107,520 images with the transcription-only objective ($\lambda=0$ in the reward). M3 (w/ UNC, SFT): apply the cold-start procedure (Sec. 5.1) to obtain 48k character-level UNC-tagged pairs, then SFT Qwen2.5-VL-7B on these (image, OCR-with-tags) pairs. M4 (w/ UNC, SFT→GRPO): continue from M3 and run GRPO on 107,520 images using the character-level reward. We use LLaMA Factory (Zheng et al., 2024) for SFT and VERL (Sheng et al., 2024) for GRPO; training hyperparameters are listed in the Appendix B.

Table 1: Results for Exp1 on Blur-OCR, with character-level cold start and character-level reward.

| Model | $\mathbf{F_1}$ (↑) | | **Accuracy** (↑) | |
|---|---|---|---|---|
| | char | word | char | word |
| base: Qwen2.5-VL-7B | - | - | 0.858 | 0.776 |
| M1: w/o UNC, SFT | - | - | 0.868 | 0.752 |
| M2: w/o UNC, SFT→GRPO | - | - | 0.892 | 0.817 |
| M3: w/ UNC, SFT | 0.243 | 0.286 | 0.830 | 0.710 |
| M4: w/ UNC, SFT→GRPO | 0.289 | 0.363 | 0.889 | 0.829 |

**Results and analysis.** From Tab. 1, we observe that the untuned base Qwen2.5-VL-7B starts at 0.858/0.776 (char/word). Adding UNC tags only during SFT reduces accuracy relative to SFT without tags (M3 vs. M1), while GRPO without tags (M2) improves over SFT. Notably, optimizing tagging with GRPO in M4 restores accuracy while also improving tagging quality, with word $F_1 = 0.363$, up from 0.286 with SFT-only tags. In summary, GRPO removes the accuracy penalty caused by SFT with tags and yields more reliable uncertainty spans, outperforming the baselines.

### 7.2 EXP2: COLD START AND REWARD GRANULARITY

**Setup.** Next, we investigate two questions: (i) whether GRPO can learn UNC tagging without a cold start and whether a random-tag cold start is sufficient, and (ii) how matching versus mismatching the granularity of the cold start and the GRPO reward affects performance. For cold starts, SFT uses 48,000 data pairs; GRPO uses 107,520 images. "Random-tag" cold start injects UNC tags at random alphanumeric positions. See Appendix E for details. "No cold start" starts GRPO directly from the base model with an instruction on how to use tags, prompt in Appendix D. We set $\lambda=0.9, \beta=1$ in the reward for this experiment.

**Results and analysis.** From Tab. 2, we observe that although GRPO improves transcription across the board, a cold start is essential for meaningful UNC usage. Without a cold start the model produces almost no tags ($F_1 \approx 0.01$), and a random start teaches only the format, with poor placement

Table 2: Exp2 on Blur-OCR: impact of cold-start (CS) supervision and reward granularity. Metrics follow Sec. 4 and "∅" denotes none for the corresponding column (i.e., no cold start or GRPO).

| CS | Reward | Precision (↑) | | Recall (↑) | | $F_1$ (↑) | | Accuracy (↑) | | Gap (↑) | |
|---|---|---|---|---|---|---|---|---|---|---|---|
| | | char | word | char | word | char | word | char | word | char | word |
| ∅ | Char | 0.013 | 0.012 | 0.007 | 0.005 | 0.010 | 0.009 | 0.870 | 0.735 | 0.000 | 0.001 |
| Random | Char | 0.207 | 0.310 | 0.226 | 0.319 | 0.210 | 0.298 | 0.884 | 0.752 | 0.190 | 0.203 |
| | Word | 0.120 | 0.248 | 0.236 | 0.285 | 0.159 | 0.269 | 0.879 | 0.820 | 0.239 | 0.258 |
| Word | ∅ | 0.198 | 0.435 | 0.266 | 0.220 | 0.200 | 0.258 | 0.811 | 0.671 | 0.175 | 0.308 |
| | Char | 0.231 | 0.434 | 0.299 | 0.231 | 0.240 | 0.300 | 0.895 | 0.830 | 0.187 | 0.317 |
| | Word | 0.205 | 0.488 | 0.404 | 0.366 | 0.225 | 0.378 | 0.879 | 0.805 | 0.162 | 0.381 |
| Char | ∅ | 0.335 | 0.524 | 0.247 | 0.233 | 0.243 | 0.286 | 0.830 | 0.710 | 0.268 | 0.396 |
| | Char | 0.342 | 0.523 | 0.306 | 0.312 | 0.289 | 0.363 | 0.889 | 0.829 | 0.310 | 0.437 |
| | Word | 0.445 | 0.683 | 0.513 | 0.549 | 0.422 | 0.574 | 0.885 | 0.839 | 0.433 | 0.590 |

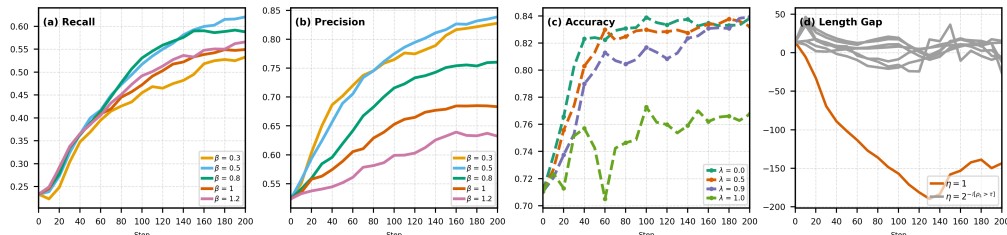

Figure 3: Exp3: (a-b) $\beta$: controls the precision–recall trade-off; $\beta \approx 0.5$ gives the fastest precision gains with stable recall, while larger $\beta$ hurts precision and very small $\beta$ slightly lowers recall. (c) $\lambda$: runs with $\lambda<1$ converge to higher accuracy than $\lambda=1$, which tends to manufacture errors. (d) $\eta$: the red curve ($\eta=1$, no damping) steadily shortens outputs, whereas the bundle of gray curves (damping on, $\eta = 2^{-\mathbb{I}[\rho_L>1.3]}$ across various $(\beta,\lambda)$) stay near zero length-gap and thus remain stable. (a-c) Word-level. (d) Char-level.

(word $F_1 \approx 0.27$–$0.29$). After applying structured CS followed by GRPO, the accuracies of different configurations are broadly similar, so the main differences lie in UNC quality. The best UNC performance is achieved by Char-CS paired with a Word-reward, yielding strong uncertainty metrics (word $F_1 \approx 0.57$, Gap $\approx 0.59$) while maintaining high accuracy. Our interpretation is that Char-CS teaches precise, tight boundaries so the model can place tags with higher precision, while the word-level reward then provides denser and more forgiving credit, so the model can accumulate effective rewards and improve steadily. In contrast, a Char-reward is too strict and provides sparse credit, which makes stable improvement more difficult. Therefore, in our subsequent experiments we use a character-level cold start with a word-level reward to train models.

### 7.3 EXP3: SENSITIVITY TO $\beta$, $\lambda$, AND $\eta$

**Setup.** Building on Exp1 and Exp2, we fix the training recipe to Char-CS + Word-reward and vary only the reward hyperparameters. We investigate the parameter ranges:

(i) $\beta \in \{0.3, 0.5, 0.8, 1.0, 1.2\}$, which controls the precision–recall balance in $F_{\beta,L}$.

(ii) $\lambda \in \{0.0, 0.5, 0.9, 1.0\}$ to probe both the tagging weight and the slope/monotonicity of the upper bound $\mathcal{R}_{\max}(e) = 1-(1-\lambda)e$: $\lambda = 0$ removes the tagging term; for $0 < \lambda < 1$ the bound decreases with error with slope $-(1-\lambda)$; $\lambda = 1$ makes it flat.

(iii) $\eta \in \{2^{-\mathbb{I}[\rho_L>1.3]}, 1.0\}$ to compare our length-mismatch damping (active only when $\rho_L > 1.3$) versus no damping.

**Results and analysis.** The results in Tab. 3 and Fig. 3 show some key trends at the word-level.

*(i) $\beta$.* Large $\beta$ drives aggressive tagging, inflating UNC spans and collapsing precision (see Fig. 3a–b). In contrast, $\beta \in [0.5, 1)$ accelerates precision without overly harming recall; the recall curves change little across $\beta$, while $\beta=1.2$ still yields poor precision with only minor recall gains. Overall,

Table 3: Exp3 on Blur-OCR with Char-CS and Word-reward, with ablation over $\beta$, $\lambda$, and $\eta$. Overall, $\beta \approx 0.5$ gives the best precision–recall balance, $\lambda<1$ maintains higher accuracy than $\lambda=1$, and turning on $\eta$ damping stabilizes length and improves accuracy.

| Ablation | Setting | Precision (↑) | | Recall (↑) | | $F_1$ (↑) | | Accuracy (↑) | | Gap (↑) | |
|---|---|---|---|---|---|---|---|---|---|---|---|
| | | char | word | char | word | char | word | char | word | char | word |
| $\lambda=0.9, \eta=2^{-\mathbb{I}[\rho_L>1.3]}$ | $\beta=0.3$ | 0.630 | 0.828 | 0.510 | 0.533 | 0.529 | 0.617 | 0.885 | 0.831 | 0.693 | 0.714 |
| | $\beta=0.5$ | 0.626 | 0.839 | 0.599 | 0.620 | 0.572 | 0.685 | 0.881 | 0.839 | 0.694 | 0.739 |
| | $\beta=0.8$ | 0.485 | 0.760 | 0.532 | 0.588 | 0.486 | 0.624 | 0.883 | 0.832 | 0.478 | 0.630 |
| | $\beta=1.0$ | 0.445 | 0.683 | 0.513 | 0.549 | 0.442 | 0.574 | 0.885 | 0.839 | 0.433 | 0.590 |
| | $\beta=1.2$ | 0.341 | 0.552 | 0.521 | 0.566 | 0.320 | 0.558 | 0.875 | 0.814 | 0.380 | 0.526 |
| $\beta=0.5, \eta=2^{-\mathbb{I}[\rho_L>1.3]}$ | $\lambda=0.0$ | 0.289 | 0.451 | 0.189 | 0.192 | 0.230 | 0.250 | 0.880 | 0.838 | 0.241 | 0.352 |
| | $\lambda=0.5$ | 0.534 | 0.628 | 0.498 | 0.514 | 0.482 | 0.581 | 0.882 | 0.832 | 0.572 | 0.625 |
| | $\lambda=0.9$ | 0.626 | 0.839 | 0.599 | 0.620 | 0.572 | 0.685 | 0.881 | 0.839 | 0.694 | 0.739 |
| | $\lambda=1.0$ | 0.555 | 0.941 | 0.748 | 0.747 | 0.591 | 0.811 | 0.814 | 0.767 | 0.585 | 0.835 |
| $\beta=0.5, \lambda=0.9$ | $\eta=2^{-\mathbb{I}[\rho_L>1.3]}$ | 0.626 | 0.839 | 0.599 | 0.620 | 0.572 | 0.685 | 0.881 | 0.839 | 0.694 | 0.739 |
| | $\eta=1.0$ | 0.600 | 0.816 | 0.510 | 0.511 | 0.486 | 0.577 | 0.792 | 0.782 | 0.523 | 0.684 |

$\beta=0.5$ gives the best balance with precision ($\approx 0.839$) and recall ($\approx 0.620$), and also there is a large error rate gap ($\approx 0.739$), so we adopt $\beta=0.5$ thereafter.

*(ii)* $\lambda$. In Fig. 3c, $\lambda=1$ raises accuracy early but then stalls and converges below the $\lambda<1$ runs. The reason is the flat upper bound $\mathcal{R}_{\max}(e) = 1-(1-\lambda)e$ at $\lambda=1$: once errors are perfectly wrapped by UNC, adding more errors no longer reduces reward, so the policy could tend to manufacture and tag errors rather than only tag necessary spans. Using a larger weight below 1 (we use $\lambda=0.9$) restores a negative slope on $e$, allowing accuracy to improve while still providing a strong tagging signal.

*(iii)* $\eta$. With no damping ($\eta=1$), the output length keeps decreasing during training (Fig. 3d), which lowers accuracy, while tagging scores rise. Since our UNC rule assigns each missing segment to the nearest UNC span, the model can shorten its output and place a few tags near boundaries. Thus, many of the missing tokens count as "inside" UNC, artificially boosting precision/recall/$F_1$ without real transcription. Notably, under $\lambda=0.9$ with no damping, a degenerate "tag-everything, say-little" strategy can still earn high reward; enabling the damping $\eta = 2^{-\mathbb{I}[\rho_L>\tau]}$ counteracts this behavior. Based on these results, we use $\beta=0.5$, $\lambda=0.9$, and the damping rule for $\eta$ in subsequent experiments, as this setting maintains strong transcription while producing precise, well-calibrated UNC tags.

## 7.4 Exp4: Cross-model comparison on *Blur-OCR*

**Setup.** We compare general VLMs and OCR-specialized models on the Blur-OCR dataset. General models receive an instruction to transcribe and wrap uncertain spans with UNC tags (see prompt in Appendix D), and decoded with temperature $0.2$. OCR baselines are run with their recommended inference settings, they cannot emit tags, so the entries for tagging metrics and Gap are marked "–". In addition, we train uncertainty-aware models based on Qwen-2.5-VL and InternVL2.5 with $\beta=0.5$ and $\lambda=0.9$ using our method and apply greedy decoding at test time. We also train a plain OCR model and derive uncertainty from token-level Shannon entropy (Manakul et al., 2023), where a single threshold is selected on validation set to maximize $F_1$. Tokens whose entropy exceeds the threshold are marked uncertain. Finally, we include a multi-model voting baseline, *Ensembles*: we run five general VLMs for transcription and take Gemini-2.5-Pro as the anchor. If at least two of the other four models disagree with Gemini on a region, then we mark that region as uncertain. We use these two conventional uncertainty quantification (UQ) methods as baselines for comparison.

**Results and analysis.** The results are in Tab. 4. General VLMs achieve similar word-level accuracy, from 0.733 to 0.826. Within this group, models that expose stepwise reasoning (Gemini-2.5-Pro, Claude-Opus-4, InternVL-3.5-241B-A28B) obtain higher word-level $F_1$ for tagging (0.156–0.205) than non-reasoning models such as GPT-4o and Qwen-2.5-VL-72B (0.039 and 0.013), although the absolute numbers remain low. The *Ensembles* approach achieves reasonable recall but suffers from low precision, while the entropy baseline provides partial localization with 0.410 word-level $F_1$. Meanwhile, OCR-specialized systems perform poorly on Blur-OCR, partly due to smaller parameter counts and limited exposure to heavily degraded pages; their word-level accuracy ranges from 0.219 to 0.732, and GOT exhibits repetition artifacts. In contrast, our uncertainty-aware Qwen-2.5-VL-7B model reaches 0.685 word-level $F_1$ with 0.839 word-level accuracy, outperforming all baselines on

Table 4: Exp4 on *Blur-OCR*: cross-model comparison.

| Category | Model | Precision (↑) | | Recall (↑) | | F₁ (↑) | | Accuracy (↑) | | Gap (↑) | |
|---|---|---|---|---|---|---|---|---|---|---|---|
| | | char | word | char | word | char | word | char | word | char | word |
| General VLMs | GPT-4o-2024-11-20 | 0.111 | 0.151 | 0.021 | 0.026 | 0.032 | 0.039 | 0.839 | 0.734 | 0.070 | -0.003 |
| | Qwen-2.5-VL-72B | 0.079 | 0.117 | 0.009 | 0.008 | 0.014 | 0.013 | 0.873 | 0.801 | 0.053 | -0.039 |
| | Gemini-2.5-Pro | 0.185 | 0.348 | 0.156 | 0.138 | 0.143 | 0.163 | 0.880 | 0.826 | 0.158 | 0.249 |
| | Claude-opus-4-20250514 | 0.305 | 0.421 | 0.176 | 0.155 | 0.187 | 0.205 | 0.840 | 0.733 | 0.275 | 0.277 |
| | InternVL-3.5-241B-A28B | 0.279 | 0.377 | 0.120 | 0.138 | 0.131 | 0.156 | 0.853 | 0.752 | 0.162 | 0.201 |
| OCR-specialized | MinerU2.5 (Niu et al., 2025) | – | – | – | – | – | – | 0.801 | 0.732 | – | – |
| | PaddleOCR 3.2.0 (Cui et al., 2025) | – | – | – | – | – | – | 0.702 | 0.308 | – | – |
| | Dolphin (Feng et al., 2025) | – | – | – | – | – | – | 0.632 | 0.549 | – | – |
| | GOT Wei et al. (2024) | – | – | – | – | – | – | 0.239 | 0.219 | – | – |
| UQ baseline | *Ensembles* | 0.389 | 0.487 | **0.703** | **0.742** | 0.458 | 0.491 | 0.880 | 0.826 | 0.358 | 0.418 |
| | Qwen-2.5-VL-7B (entropy) | 0.327 | 0.406 | 0.339 | 0.440 | 0.313 | 0.410 | **0.888** | **0.840** | 0.304 | 0.350 |
| Ours | InternVL2.5-8B (UNC) | **0.669** | 0.802 | 0.535 | 0.579 | 0.550 | 0.622 | 0.849 | 0.748 | 0.683 | 0.700 |
| | Qwen-2.5-VL-3B (UNC) | 0.548 | 0.786 | 0.520 | 0.613 | 0.504 | 0.638 | 0.872 | 0.798 | 0.623 | 0.636 |
| | Qwen-2.5-VL-7B (UNC) | 0.626 | **0.839** | 0.599 | 0.620 | **0.572** | **0.685** | 0.881 | 0.839 | **0.694** | **0.739** |

uncertainty localization while preserving strong transcription. Our uncertainty-aware InternVL2.5 model learns a similar ability to bracket uncertain spans, indicating that the approach transfers across architectures. Detailed qualitative examples are provided in Appendix F.

## 8 CONCLUSION AND FUTURE WORK

This work introduced an uncertainty-aware OCR approach that trains VLM-based OCR systems to bracket self-identified uncertain spans with uncertainty tags on degraded documents. Using a pseudo-labeled cold start and a GRPO-driven multi-objective reward, our method preserves transcription quality while accurately localizing uncertainty. We also introduced Blur-OCR, a challenging OCR benchmark consisting of document images with highly diverse synthetic degradations, comprising 107,520 training images and 2,048 evaluation images. On the Blur-OCR benchmark, our best 7B model with Char-CS + Word-reward achieves a word-level F1 of 0.685, outperforming both open- and closed-source baselines for uncertainty tagging without loss of accuracy. These results demonstrate that making uncertainty explicit offers a practical path to more trustworthy degraded document understanding.

While our experiments instantiate this framework in degraded document OCR, the underlying problem is more general. Our goal is to train VLMs to explicitly express uncertainty when their inputs are difficult or ambiguous, for example when the visual evidence is lossy or compressed, or when the underlying content is intrinsically hard to recognize, rather than guessing from incomplete information. Similar challenges arise in many other settings. For example, in speech recognition from noisy or low-quality audio, an ASR-LLM system (Bai et al., 2024) could mark acoustically ambiguous regions with UNC tags. Recent work such as DeepSeek-OCR (Wei et al., 2025) compresses long visual contexts into visual tokens, which act as a low-fidelity sketch of the original document. This compression is analogous to our degraded images, and UNC tags could similarly flag where the sketch loses critical detail. QA systems that must answer under partial evidence could also benefit from localized uncertainty marking rather than overconfident generation. Another direction for future work is to train on more diverse document distributions to build a general-purpose OCR system. We also see our proposed methods as a starting point for broader uncertainty-aware generation from lossy inputs, generalizing beyond OCR.

## 9 REPRODUCIBILITY STATEMENT

The paper specifies the tagging semantics and evaluation protocol in Sections 3 and 4, the pseudo-labeled cold start in Section 5.1, and the GRPO objective in Section 5.2. Experimental settings are detailed in Section 7; training hardware and key hyperparameters are listed in Appendix B; prompts are provided in Appendix D; and UNC tagging examples are given in Appendix C.

Our training pipeline builds on open-source frameworks. We release code and resources at `https://github.com/NikoGuan/Uncertainty_OCR`, including implementations for generating pseudo-label UNC tags from model-ground-truth alignments and computing the GRPO reward. We also provide the Blur-OCR benchmark dataset.

## ACKNOWLEDGMENTS

This publication is part of a project that has received funding from (i) the European Research Council (ERC) under the Horizon 2020 research and innovation programme (Grant agreement No. 884951); (ii) Research Ireland to the Insight Centre for Data Analytics under grant No. 12/RC/2289_P2.

This publication is also based on research funded by European Union's Horizon Europe 2021–2027 framework programme, Marie Skłodowska-Curie Actions, Grant Agreement No. 101072456; Taighde Éireann – Research Ireland under grant No. 13/RC/2094_2 to Lero the Research Ireland Centre for Software.

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

## A    USE OF LLMS

We used LLMs as general-purpose assistive tools for literature search and language polishing. We also trained and evaluated LLMs on the Blur-OCR benchmark.

## B    TRAINING HYPERPARAMETERS

The following describes our training configuration. All runs were on 8× NVIDIA H200 (141 GB) with bf16. For SFT (LLaMA Factory), we used per-device batch size = 1 with gradient accumulation = 4 (effective batch size = 32 on 8 GPUs), AdamW, learning rate $2.5\times10^{-6}$, cosine schedule with 10% warmup, and 2 epochs; other SFT options were left at library defaults. For GRPO (VERL), we trained 1 epoch with global train batch size 512, actor learning rate $4.0\times10^{-6}$ and a cosine warmup (warmup ratio 0.03, min-LR ratio 0.10, cycles 0.5), entropy coeff 0.001, and we set $\beta_{KL} = 0$. Rollouts used bf16 on a vLLM backend and sampled 5 completions per input. All other engine/FS-DP/offload/caching/prefill flags followed the framework defaults. The 2,160 SFT steps took around 1 hour. GRPO 210 steps took around 20 hours.

## C    UNC TAGGING EXAMPLES

Here we illustrate how pseudo-labels are constructed for our proposed cold start strategy. For each degraded image, we run the base model to obtain a transcript, align it with the ground truth, and insert UNC tags into the model's transcript around localized errors. Below we show both character- and word-level variants used for SFT.

```
GT     The quick brown fox jumps
OCR    Te quick brownnn fox jumaps
Char   <C>Te</C> quick brown<C>nn</C> fox jum<C>a</C>ps
Word   <C>Te</C> quick <C>brownnn</C> fox <C>jumaps</C>

GT     The quick brown fox jumps
OCR    the qaick brown fox jumps
Char   <C>t</C>he q<C>a</C>ick brown fox jumps
Word   <C>the qaick</C> brown fox jumps

GT     hello world
OCR    hello wlrld 123
Char   hello w<C>l</C>rld<C> 123</C>
Word   hello <C>wlrld 123</C>
```

## D    PROMPT

We use the following instruction for general VLMs in Exp4:

```
You are an intelligent OCR system.
Perform OCR to transcribe all text from the image.
When you encounter characters or words you are not confident
about, wrap them with <C>...</C>.
You may use <C>...</C> multiple times.  Do not nest or
overlap <C> regions.
Use the smallest span that covers the uncertainty.  Preserve
natural reading order and basic line breaks.  Keep the
```

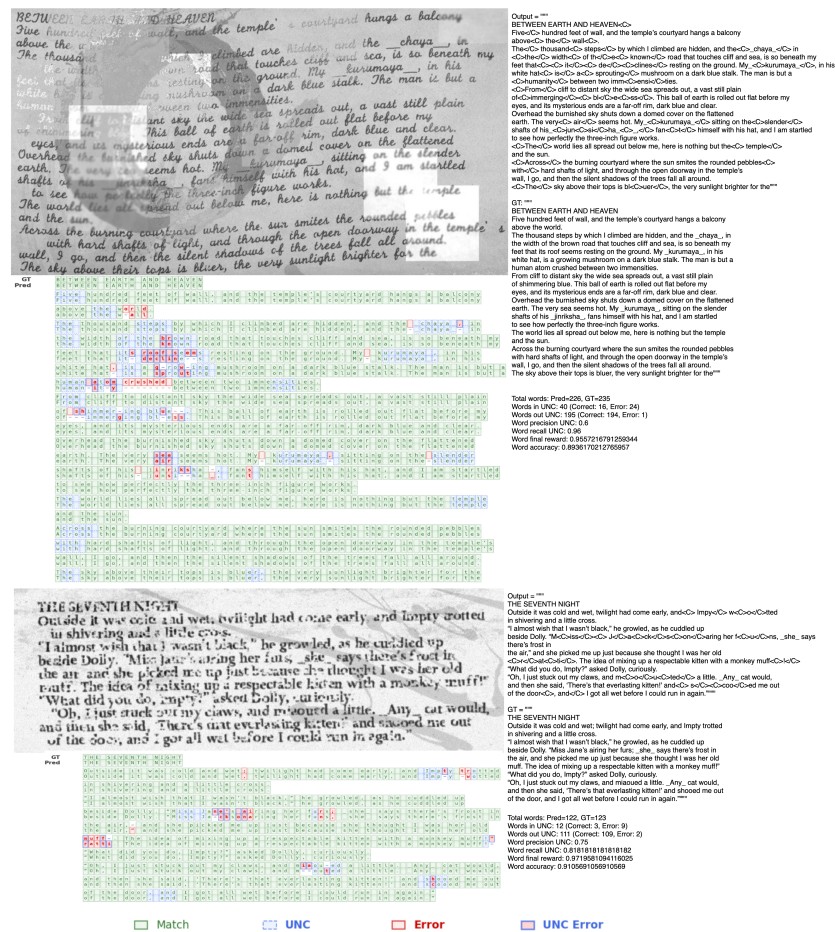

Figure 4: Examples of outputs from our uncertainty-aware OCR method on the *Blur-OCR* benchmark. For each page we show (top→bottom): the degraded input image; the model transcription with UNC tags and the ground truth (right, "Output/GT"); and an alignment map. Colors: green = match, blue = text inside UNC, red = error outside UNC, red in blue box = error inside UNC. The panel at lower right reports word-level counts (Correct/Error in/out of UNC) and summary metrics (precision/recall/$F_1$ for UNC and word accuracy).

```
original language, punctuation, and casing.  Output only
the final text (no explanations, no code blocks, no quotes).

Input:  <image>
```

## E  RANDOM-TAG COLD START (FOR EXP2)

This baseline ignores the image and inserts UNC tags at random. For each clean sentence, we split it into $S \in \{1, \ldots, 8\}$ segments. For each segment, we sample an uncertainty rate $r$ from a mixture: 10% with $r = 0$; 40% with $r \sim U[0, 0.1]$; 10% with $r \sim U[0.2, 0.5]$; 25% with $r \sim U[0.5, 0.8]$; and 15% with $r \sim U[0.8, 1.0]$. We then mark $\lceil r \cdot \text{len(segment)} \rceil$ characters in that segment as uncertain and wrap contiguous selections with <C>...</C>. This yields a non-uniform, segment-wise UNC distribution similar to real text.

## F  SAMPLE OUTPUT

This appendix provides additional qualitative examples from the *Blur-OCR* benchmark, following the visualization format in Fig. 4. For a given page, we show: (1) the degraded input image; (2) the model transcription with UNC tags alongside the ground truth (*Output/GT*); and (3) an alignment map.

Table 5: List of document degradation parameters by noise level.

| Parameter | Level-1 | Level-2 | Level-3 | Level-4 |
|---|---|---|---|---|
| Noise Factor | [0,10] | [0,30] | [0,50] | [0,50] |
| Scale Factor | [0.2,1] | [0.2,1] | [0.2,1] | [0.2,1] |
| Gaussian Blur (px) | [0,1] | [0,1] | [0,2] | [0,2] |
| Background Intensity | [0,0.1] | [0,0.3] | [0,0.6] | [0,0.6] |
| Stain Transparency | [0,0.3] | [0,0.6] | [0,0.8] | [0,0.8] |
| Max Stains | [0,1] | [0,3] | [0,5] | [0,5] |
| Contrast Factor | [0.6,1] | [0.6,1] | [0.6,1] | [0.3,1] |
| Black Spot Size (px) | 1×1 | 1×1 | 1×1 | 1×1 |
| Black Spots per Page | [0,HW/3000] | [0,HW/2000] | [0,HW/1000] | [0,HW/1000] |
| White Patch Size (px) | [0,3]×[0,3] | [0,5]×[0,5] | [0,5]×[0,5] | [0,5]×[0,5] |
| White Patches per Page | [0,HW/500] | [0,HW/300] | [0,HW/200] | [0,HW/100] |
| Line Artifacts | [0,4] | [0,6] | [0,8] | [0,10] |
| Dilation Iterations | [0,2] | [0,2] | [0,2] | [0,2] |
| Erosion Iterations | [0,2] | [0,2] | [0,2] | [0,2] |

# G DEGRADATION OPERATIONS AND PARAMETERS

To add further diversity to the degradation process, we implement four progressive degradation levels, with the corresponding parameter ranges shown in Table 5. Each level consists of a series of degradation operations applied in a random order, so that different operation sequences can produce substantially different effects. Higher levels introduce more aggressive distortions.

