# OpenReview forum: "Teaching VLMs to Admit Uncertainty in OCR from Lossy Visual Inputs"
_ICLR.cc/2026/Conference — ICLR 2026 Poster_

### Official Review · Reviewer_ixBu · 2025-10-26

**Soundness:** 2
**Presentation:** 2
**Contribution:** 2
**Rating:** 4
**Confidence:** 4

**Summary:**

This paper introduces uncertainty-aware OCR, where vision-language models transcribe degraded documents while bracketing uncertain spans with explicit uncertainty tags (`<C>...</C>`). The authors employ a pseudo-labeled cold start followed by Group Relative Policy Optimization (GRPO) with a multi-objective reward that balances transcription accuracy and uncertainty coverage. They introduce Blur-OCR, a benchmark of 2,048 synthetically degraded images from Project Gutenberg. The best model (Qwen2.5-VL-7B) achieves word-level F1 of 0.685 for uncertainty tagging and 0.839 accuracy, outperforming several baseline models including GPT-4o and Claude-Opus4.

However, the limited evaluation scope, missing key baselines, and conceptual concerns about the cold start procedure prevent a stronger recommendation. With revisions addressing the generalization questions and including comparisons with MinerU systems and uncertainty quantification methods, this could become a solid contribution.

**Strengths:**

**Clear problem formulation**: The paper addresses a real problem—VLM-based OCR systems hallucinate on degraded documents without signaling uncertainty, which is worse than classical OCR systems that produce obviously garbled output. The motivation is well-articulated.

**Systematic methodology**: The two-stage training approach (pseudo-labeled cold start + GRPO) is reasonable and well-described. The multi-objective reward design with safeguards against reward hacking (especially the length-mismatch damping factor η) demonstrates careful engineering.

**Weaknesses:**

### 1. Limited Benchmark Coverage and Missing Baselines

The evaluation is restricted to a single synthetic benchmark (Blur-OCR). The paper does not evaluate on:
- Established document understanding benchmarks like OmniDocBench [1], which provides diverse real-world PDF documents with comprehensive annotations
- More general OCR benchmarks beyond the two mentioned in Related Work (OCRBench/OCRBench v2)
- Recent document parsing systems like MinerU [2] or MinerU2.5 [3], which represent state-of-the-art in document content extraction

### 2. Incomplete Related Work on Uncertainty Quantification

The paper misses critical recent work on OCR uncertainty estimation. Notably, it does not cite or compare with methods that provide quantitative uncertainty measures. For instance, recent work on consensus entropy for multi-VLM agreement [4] provides token-level uncertainty scores that can be directly compared with the proposed tagging approach. While the paper mentions entropy-based baselines briefly in Exp4 (Section 7.4), it lacks:
- Proper contextualization within the broader uncertainty quantification or calibration literature
- Discussion of how the proposed explicit tagging approach differs from or improves upon probabilistic uncertainty measures

### 3. Conceptual Issues with the Cold Start Procedure

The pseudo-labeling strategy has a fundamental problem: it tags the **model's own errors** on degraded images, not necessarily the **visually unreadable regions**. This conflates two distinct phenomena:

a) Text that is visually degraded/unreadable in the image
b) Text where the model happened to make a mistake

The paper claims (Section 5.1) that "When the image is unreadable, models tend to guess and are often wrong," but this assumption is not validated. Two problematic cases arise:

- **False positive tags**: The model may correctly transcribe text from a degraded-but-readable region, yet the cold start labels it as uncertain simply because it differs from GT due to actual degradation differences
- **False negative tags**: The model may confidently hallucinate on clear, undegraded regions (a known VLM failure mode [5]), which would not receive uncertainty tags


### 4. Limited Analysis of Generalization Beyond Synthetic Degradations

VLMs are known to hallucinate even on clean, high-quality document images [5]. The paper does not:
- Test whether the uncertainty-aware model can identify hallucinations on non-degraded documents
- Evaluate on real-world degraded documents (e.g., historical documents, which are mentioned in the motivation but never tested)
- Compare performance on different types of errors: character substitutions vs. hallucinated words/phrases

### 5. Benchmark Construction Concerns

The Blur-OCR benchmark applies random combinations of degradations to clean text, but:
- No analysis is provided on whether the degradations are realistic compared to actual historical documents or real-world low-quality scans
- The paper does not discuss the distribution of degradation severity or provide statistics on what fraction of text becomes truly unreadable
- Figure 2 shows sample pages, but there is no quantitative analysis of degradation characteristics

This makes it difficult to assess whether Blur-OCR represents realistic use cases or is primarily useful for evaluating this specific training paradigm.

**Questions:**

1. **Generalization to real degradations**: Can the authors evaluate on real-world degraded documents (e.g., historical document datasets) to demonstrate that the approach generalizes beyond the specific synthetic degradation pipeline?

2. **Comparison with MinerU systems**: MinerU and MinerU2.5 [2,3] represent recent advances in document parsing. How does the proposed method compare against these systems on Blur-OCR? If these systems cannot produce uncertainty estimates, can they be combined with the proposed tagging approach?

3. **OmniDocBench evaluation**: OmniDocBench [1] provides diverse document types. Can the authors evaluate whether uncertainty tagging helps on this more realistic benchmark?

4. **Quantitative uncertainty measures**: How does explicit binary tagging compare with continuous uncertainty scores (e.g., consensus entropy [4], token-level entropy, or multi-model voting confidence)? Can the authors provide experiments showing when discrete tags are preferable to probabilistic scores?

5. **Hallucination on clean images**: Does the trained model successfully tag hallucinations that occur on non-degraded, high-quality document images? This would demonstrate that the model learns genuine uncertainty rather than merely memorizing the training degradation distribution.

6. **Cold start validation**: Can the authors provide analysis showing that the pseudo-labels in the cold start actually correspond to visually degraded regions? For example, human annotations on a sample of cold-start labels, or correlation analysis between degradation strength and tagging frequency.

7. **Breakdown by error type**: What types of errors are well-covered by UNC tags vs. those that escape detection? Are character-level substitutions easier to catch than word-level hallucinations?


## References

[1] Linke Ouyang, Yuan Qu, Hongbin Zhou, et al. OmniDocBench: Benchmarking Diverse PDF Document Parsing with Comprehensive Annotations. arXiv:2412.07626, 2024. (CVPR’25)

[2] Bin Wang, Chao Xu, Xiaomeng Zhao, et al. MinerU: An Open-Source Solution for Precise Document Content Extraction. arXiv:2409.18839, 2024.

[3] Junbo Niu, Zheng Liu, Zhuangcheng Gu, et al. MinerU2.5: A Decoupled Vision-Language Model for Efficient High-Resolution Document Parsing. arXiv:2509.22186, 2025.

[4] Consensus Entropy: Harnessing Multi-VLM Agreement for Self-Verifying and Self-Improving OCR

[5] Adam Tauman Kalai, Ofir Nachum, Santosh S. Vempala, and Edwin Zhang. Why language models hallucinate. arXiv:2509.04664, 2025.

---

> ### Author Response · Authors · 2025-11-19
> **Rebuttal to Reviewer ixBu (Part 1)**
>
> We thank Reviewer ixBu for the detailed review. Below we respond to the comments and would be happy to further discuss them.
>
> > **W1. Limited Benchmark Coverage and Missing Baselines.**
>
> The benchmarks mentioned by the reviewer (OmniDocBench, OCRBench/OCRBench v2) primarily target improving or evaluating extraction accuracy on clean, readable documents, where hallucinations on clean images are measured indirectly through accuracy drops, so the main metric of interest is overall accuracy. In contrast, our main contribution is orthogonal: we study whether a model can (i) maintain OCR accuracy and (ii) explicitly mark uncertain spans with UNC tags when it is unsure. Accordingly, we evaluate both UNC-tag F1 and OCR accuracy, and under comparable accuracy we place more emphasis on UNC-tag F1. Our goal is not to improve plain text recognition accuracy or document-structure extraction per se, so directly evaluating on OmniDocBench/OCRBench/OCRBench v2 would not meaningfully test the uncertainty-tagging objective which is the primary focus of the paper.
>
> Regarding MinerU/MinerU2.5: these systems, like PaddleOCR/Dolphin/GOT, achieve strong OCR performance on clean documents and are representative SOTA document parsers, but they degrade substantially on our heavily corrupted Blur-OCR images. In the revised paper (see Exp4), we include MinerU2.5 as an additional baseline. On Blur-OCR, MinerU2.5 reaches only 0.801 character-level accuracy (vs. 0.881 for our backbone + UNC), confirming that state-of-the-art clean-document parsers struggle in this regime, likely because of their smaller parameter counts or the lack of highly noisy images in their training data.
>
>
> > **W2. Incomplete Related Work on Uncertainty Quantification.**
>
> In our initial submission (Exp. 4), we already included two standard uncertainty-quantification baselines: The first is based on token-level Shannon entropy, and the second is analogous to a “multi-VLM agreement” approach, reported as “Ensembles” in Exp. 4, where we run five general LLMs and estimate uncertainty from the disagreement among their outputs.
>
> On Blur-OCR, both baselines are clearly weaker than our approach (e.g., entropy: 0.41, ensemble: 0.49, ours: 0.69). Existing methods in the literature are somewhat limited in that they are not trainable end-to-end and cannot be directly improved through task-specific training.
>
>
> >**W3. Conceptual Issues with the Cold Start Procedure.**
>
> We would like to clarify the rationale behind our Cold Start design. We claim that “when the image is unreadable, models tend to guess and are often wrong,” but we do not claim that all model errors come from unreadable regions. In the cold start stage, we specifically aim to tag the model's incorrect outputs rather than just visually degraded regions. This is because in degraded documents, the vast majority of model errors are triggered by visual ambiguity. Training the model to predict its own errors serves as a highly effective proxy for detecting visual unreadability.
>
> Furthermore, we specifically do not aim to tag all visually degraded regions. Since LLMs rely on both visual evidence and textual context, a model may confidently infer the correct content from context even in visually unreadable regions. In such cases, we do not want the model to add an UNC tag, as the output is correct and useful. Consequently, we define "uncertainty" based on the model's actual behavior rather than the noise parameters. Since the high randomness in our degradation pipeline makes it difficult to define "objective" unreadability, injecting Cold Start labels based on exact mismatches is the most robust strategy. Since these tags are derived directly from Ground Truth alignment, they strictly represent actual prediction errors, eliminating false positives or negatives in this context. Our ultimate goal is that the content inside UNC tags should be unreliable. The purpose of cold start is to teach the model how to use UNC tags; subsequent GRPO training then teaches the model how to trade off whether to attach UNC tags when it is guessing.

---

> ### Author Response · Authors · 2025-11-19
> **Rebuttal to Reviewer ixBu (Part 2)**
>
> >**W4. Identify hallucinations on non-degraded documents? Generalization Beyond Synthetic Degradations?**
>
> We agree that identifying hallucinations on high quality pages without degradation is important, but they correspond to a different failure mode from the one we target. Our method is designed for input-driven uncertainty, where the visual evidence itself is lossy or corrupted. On the Blur-OCR benchmark, errors concentrate in visually ambiguous regions, and our cold-start + GRPO training explicitly teaches the model to tag spans whose underlying evidence is unreliable. In contrast, hallucinations on clean images typically arise when the model relies more on its learned text-generation prior than on the actual visual evidence, which would require different supervision and is beyond the scope of this work.
>
> We also agree that evaluation on real-world degradations is important. However, we are not aware of any large-scale corpus of real degraded documents with high-quality ground-truth transcripts, which motivated us to construct Blur-OCR using synthetic degradations. Prior work (e.g., PreP-OCR) has shown that similar synthetic degradation pipelines can transfer effectively to real degraded documents. In addition, we qualitatively evaluated our UNC model on a small set of real degraded documents and observed broadly similar UNC behavior to that on Blur-OCR, although we do not have ground-truth of degradation regions and therefore cannot provide a detailed quantitative analysis.
>
> > **W5. Benchmark Construction Concerns.**
>
> We appreciate the reviewer’s suggestion. Blur-OCR is intentionally constructed as a challenging OCR benchmark. Unlike prior benchmarks that mostly contain clean text, it is specifically designed to test whether large models can admit uncertainty when parts of the visual evidence are genuinely damaged, rather than always producing fully confident guesses. As long as some regions are indeed visually degraded and the degradation types are sufficiently diverse, the benchmark is meaningful for evaluating this uncertainty-tagging behavior.
>
> The reviewer asks what fraction of text becomes “truly unreadable.” In our view, this notion is inherently model- and observer-dependent: different LLMs or human readers may disagree on which regions are still decipherable. Therefore, we do not attempt to explicitly annotate which characters are “unreadable” in each image. Instead, to help readers better understand the degradation characteristics, in the revised version of the paper we add further description of Blur-OCR (degradation types and parameter ranges) in Section 6 and the Appendix.
>
> > **Q1. Generalization to real degradations: Can the authors evaluate on real-world degraded documents (e.g., historical document datasets) to demonstrate that the approach generalizes beyond the specific synthetic degradation pipeline?**
>
> See **W4** for a detailed answer.
>
> > **Q2. Comparison with MinerU systems: MinerU and MinerU2.5 represent recent advances in document parsing. How does the proposed method compare against these systems on Blur-OCR? If these systems cannot produce uncertainty estimates, can they be combined with the proposed tagging approach?**
>
> MinerU2.5 provides very strong document parsing on clean documents, but the recognition accuracy degrades substantially on noisy images. On Blur-OCR, MinerU2.5 achieves only 0.801 character-level accuracy (see W1 for details), which is significantly lower than our backbone model and general LLMs in this heavily degraded setting.
>
> In principle, MinerU2.5 can be combined with our proposed tagging approach. Our method is applied on top of a backbone recognizer to add an uncertainty-aware behavior, so MinerU2.5 could likewise be augmented with UNC tagging. However, tagging uncertainty is a relatively complex capability, and the smaller parameter count of MinerU2.5 may limit how well this behavior can be learned. We view systematically exploring such combinations as an interesting direction for future work.

---

> > ### Author Response · Authors · 2025-11-19
> > **Rebuttal to Reviewer ixBu (Part 3)**
> >
> > > **Q3. OmniDocBench evaluation: OmniDocBench provides diverse document types. Can the authors evaluate whether uncertainty tagging helps on this more realistic benchmark?**
> >
> > OmniDocBench indeed provides diverse document types, but its design goal is fundamentally different from ours, see **W1** for more detail. In this paper we restrict ourselves to content recognition to validate the feasibility of the proposed UNC mechanism. Extending this to diverse, multi-layout documents is outside the scope of this paper, but represents an interesting direction for future work.
> >
> > > **Q4. Quantitative uncertainty measures: How does explicit binary tagging compare with continuous uncertainty scores (e.g., consensus entropy, token-level entropy, or multi-model voting confidence)? Can the authors provide experiments showing when discrete tags are preferable to probabilistic scores?**
> >
> > In our initial submission (Exp. 4), we already included two standard uncertainty-quantification baselines. See **W2** for detail.
> > > **Q5. Hallucination on clean images: Does the trained model successfully tag hallucinations that occur on non-degraded, high-quality document images? This would demonstrate that the model learns genuine uncertainty rather than merely memorizing the training degradation distribution.**
> >
> > Hallucination on clean images corresponds to a different failure mode from the one we target. Our method is designed for input-driven uncertainty, where the visual evidence itself is lossy or corrupted. On Blur-OCR, errors are concentrated in visually ambiguous regions. Our cold-start + GRPO training explicitly teaches the model to tag spans whose underlying evidence is unreliable. In contrast, hallucinations on clean images typically arise when the model relies more on its learned text-generation prior than on the actual visual evidence, which would require different supervision and is beyond the scope of this work.
> >
> > > **Q6. Cold start validation: Can the authors provide analysis showing that the pseudo-labels in the cold start actually correspond to visually degraded regions? For example, human annotations on a sample of cold-start labels, or correlation analysis between degradation strength and tagging frequency.**
> >
> > We do not claim that the pseudo-labels in the cold start actually correspond to visually degraded regions. Please see **W3** for a detailed answer.
> >
> > > **Q7. Breakdown by error type: What types of errors are well-covered by UNC tags vs. those that escape detection? Are character-level substitutions easier to catch than word-level hallucinations?**
> >
> > In each experiment, we report UNC-tag recall, which directly reflects how many erroneous spans are well-covered versus how many escape detection. From these metrics, we observe that character-level errors are generally more difficult to detect comprehensively, as indicated by consistently higher word-level recall compared to character-level recall.

---

> > > ### Comment · Reviewer_ixBu · 2025-11-20
> > > **make this work more solid and professional**
> > >
> > > Thank you for the response. Some of my concerns have been addressed. I have also reviewed the responses from other reviewers, and I acknowledge the value of the current task, considering the widespread visually degraded documents issues that occur during LLM pretraining data processing.
> > >
> > > However, I would like to draw attention to the following issue:
> > > The subject discussed in the paper that possesses OCR capabilities is Large language models (LLMs) with **visual capabilities**. This term generally refers to VLM, MLLM, or LVLM. However, LLM absolutely does not refer to (image, textual prompt) to text, but rather text to text. From this perspective, the paper title is quite unprofessional. This topic should be about vision language models on degraded documents. This is also why I initially gave a negative score. I find it difficult to convince myself that work that gets this concept wrong would be accepted by the ICLR community. I hope the authors' attention to this issue in this round of discussion goes beyond merely stating `Thank you for the suggestion. We will consider using the term "VLM."`
> > >
> > > This title and the authors' expression have also led to widespread misunderstandings. The scope of uncertainty should be defined more pragmatically.
> > >
> > > The uncertainties arising from various real-world scenarios such as handwritten text, text positioning, language mixing, and model visual recognition hallucinations cannot be identified and quantified by this work. In fact, what this work addresses is:
> > > **Identifying which parts of document images are degraded** during the OCR process.
> > > Therefore, the current contribution of presentation is somewhat overclaimed.
> > >
> > > Given that this work does effectively solve problems within this scope, I suggest the authors revise the wording in this work to make the paper more professional and avoid widespread misunderstandings.
> > >
> > > After including uncertainty quantification methods in the comparison, this work will appear somewhat awkward in Table 4, because in fact `multi VLM ensembles` and `entropy corresponding to output token logits` perform quite well. I recommend plotting token lengths vs Cumulative Score or computational cost vs Cumulative Score as in Figure 5 of [1]. These two methods may achieve better final performance, but they are weaker than GRPO-trained VLMs in terms of token overhead, since this overhead should be equivalent to a single model.
> > >
> > > Furthermore, precisely because this work only addresses `identifying which parts of document images are degraded during the OCR process`, it may perform weakly on the benchmarks I suggested for comparison, as they are too clean. However, I think the authors could retrieve subsets with occlusion and other issues from these benchmarks and compare the results. The value of this comparison is that it can **alleviate concerns about the generalization of the work after fixing the overclaimed scope of uncertainty**, considering that the blur-OCR dataset has a small sample size and relatively homogeneous content. Based on my understanding of GRPO, it should be able to generalize to real-world document extraction scenarios with occlusion issues.
> > >
> > > I hope the authors will take these issues seriously and make this work more solid and professional.

---

> > > > ### Author Response · Authors · 2025-11-20
> > > > **Rebuttal to Reviewer ixBu (Part 4)**
> > > >
> > > > We are pleased that our previous responses have addressed many of the reviewer’s concerns, and we appreciate their continued engagement and acknowledgment of the value of the current task. Below we respond to the new comments point by point. We hope that our clarifications and additional analysis address the reviewer’s remaining concerns and will be helpful for the final evaluation.
> > > >
> > > >
> > > > > **C1. LLM absolutely does not refer to (image, textual prompt) to text, but rather text to text. From this perspective, the paper title is quite unprofessional.**
> > > >
> > > > We appreciate the reviewer’s careful attention to terminology. In industrial practice, the term “LLM” is often used broadly to include models with visual capabilities (e.g., ChatGPT, Gemini, etc.), and this was the informal sense in which we originally used “LLM with visual capabilities.” However, we agree that in the current context it is clearer to use “VLM.” We have updated the revised PDF accordingly. In the revised version, we have therefore updated the title and main text to use “VLM” rather than “LLM.”
> > > >
> > > >
> > > > >**C2. The uncertainties arising from various real-world scenarios such as handwritten text, text positioning etc. What this work addresses is: Identifying which parts of document images are degraded during the OCR process. Therefore, the current contribution of presentation is somewhat overclaimed.**
> > > >
> > > > We would like to clarify that our work does **not** aim to “identify which parts of the document image are degraded.” Instead, our objective is to teach the model, given both the visual input and the textual context, to recognize when its output is unreliable and to explicitly bracket those spans with UNC tags, this is the behavior we want for trustworthy OCR. This is the definition of “uncertainty” which we also discussed in our earlier response (**W3**).
> > > >
> > > > If our goal were truly to detect which parts of the image are degraded, this would be a different task (i.e., a localization / detection problem) and would call for a different type of model.
> > > >
> > > > We also note that the same training recipe can be applied to the scenarios mentioned by the reviewer. For example, in the case of messy handwritten text, where the model frequently makes transcription errors, we can similarly train it to mark those high-error regions in its output with UNC tags. Thus, while our experiments in this paper focus on degradation-induced uncertainty, the method itself is not limited to this particular source of errors.
> > > >
> > > > For these reasons, we respectfully do not believe that the contribution is overclaimed.
> > > >
> > > >
> > > > >**C3. After including uncertainty quantification methods in the comparison, this work will appear somewhat awkward in Table 4, because in fact multi VLM ensembles and entropy corresponding to output token logits perform quite well. I recommend plotting token lengths vs Cumulative Score or computational cost vs Cumulative Score as in Figure 5 of [1]. These two methods may achieve better final performance, but they are weaker than GRPO-trained VLMs in terms of token overhead, since this overhead should be equivalent to a single model.**
> > > >
> > > > We thank the reviewer for raising this point and for suggesting a more detailed cost-performance analysis. On Blur-OCR, however, our current results indicate that the entropy and multi-VLM ensemble **baselines do not perform “quite well”** compared to our method in terms of uncertainty tagging:
> > > >
> > > > - Entropy baseline: F1 = 0.410, precision = 0.406
> > > > - Ensemble baseline: F1 = 0.491, precision = 0.487
> > > > - Ours (GRPO-trained UNC tagging): F1 = 0.685, precision = 0.839
> > > >
> > > > Thus, our approach improves F1 by a large margin and nearly doubles precision relative to both non-trainable baselines. Based on these numbers, we do not consider entropy or ensembles to be competitive in this setting.
> > > >
> > > > Regarding the reviewer’s pointer to Figure 5 in reference [1], we were not able to locate a Figure 5 that exactly matches the reviewer’s description, and we suspect the intended pointer might be to reference [4]. But as the reviewer points out, our method uses token overhead equivalent to a single model while achieving better F1 than both baselines, so we expect that an additional cost-performance plot would mainly reinforce the conclusions already supported by Table 4.

---

> > > > > ### Author Response · Authors · 2025-11-20
> > > > > **Rebuttal to Reviewer ixBu (Part 5)**
> > > > >
> > > > > >**C4. Furthermore, precisely because this work only addresses “identifying which parts of document images are degraded during the OCR process”, it may perform weakly on the benchmarks I suggested for comparison, as they are too clean. However, I think the authors could retrieve subsets with occlusion and other issues from these benchmarks and compare the results. The value of this comparison is that it can alleviate concerns about the generalization of the work after fixing the overclaimed scope of uncertainty, considering that the blur-OCR dataset has a small sample size and relatively homogeneous content. Based on my understanding of GRPO, it should be able to generalize to real-world document extraction scenarios with occlusion issues.**
> > > > >
> > > > > First, as clarified in our response to **C2**, our work does not only address “identifying which parts of document images are degraded during the OCR process.” Our goal is to train the model to mark those output spans that are likely to be unreliable when the underlying visual evidence is lossy or corrupted, rather than to explicitly segment degraded regions which exist in the original image itself.
> > > > >
> > > > > In standard text recognition settings where the input is clean and the content is simple, we would not expect the model to express much uncertainty. As we discussed in **W1**, benchmarks such as OmniDocBench and OCRBench/OCRBench v2 are primarily designed to evaluate recognition and extraction accuracy on clean documents, where hallucinations on non-degraded inputs are measured indirectly via accuracy drops. This is a different failure mode from the one we target in this paper, which focuses on hallucinations driven by lossy visual evidence. In **W4**, we also highlighted the conceptual distinction between hallucinations on clean inputs and those on lossy inputs. Our GRPO objective explicitly encourages the model to avoid UNC tags when it is confident, so the same model is applied uniformly to both clean and degraded images.
> > > > >
> > > > > We appreciate the reviewer’s suggestion to retrieve subsets with occlusions and other issues from existing benchmarks. In the current review cycle, systematically retrieving and curating such subsets for span-level UNC evaluation would be challenging in terms of time and annotation effort, so we have not included these experiments in this version. We fully agree, however, that testing on more realistic occlusion scenarios would be very valuable, and we view this as a natural extension of our work. Since GRPO is agnostic to the specific source of information loss, we also share the reviewer’s intuition that our UNC-tagging mechanism should be able to generalize to real-world document extraction scenarios involving occlusions and related artifacts.

---

> > > > > > ### Comment · Reviewer_ixBu · 2025-11-21
> > > > > > **The authors' response regarding data coverage and evaluation costs is not convincing.**
> > > > > >
> > > > > > 1. On Data Coverage & Overclaiming:
> > > > > > It is logically impossible for a training set derived solely from degraded text to cover all the uncertainty types I previously mentioned.
> > > > > > Please provide a one-sentence explanation for this specific case: When strong models (e.g., InternVL) fail to recognize complex math formulas, how does your current data construction pipeline ensure that "uncertainty tags" are generated to correctly enclose these missing formulas?
> > > > > > This issue represents the challenge of "high-density formula data" in OCR uncertainty. I suspect this scenario was not fully considered in your design, which likely explains why benchmarks like OmniDocBench were excluded.
> > > > > > **I acknowledge the novelty of your method in handling degraded documents, but I strongly advise the authors to limit their claims to this specific scope.** The OCR field is vast, and there are always uncertainty types and cases beyond simple degradation. Overclaiming that your method solves "all" uncertainty types sets a misleading baseline and **creates unnecessary hurdles for future follow-up works.**
> > > > > > 2. On Evaluation Cost:
> > > > > > The argument that testing on new datasets is too costly is invalid. Evaluation on a new benchmark requires only inference, not re-training. The computational cost is negligible and fully acceptable—unless the underlying GRPO training process is unstable.

---

> > > > > > > ### Author Response · Authors · 2025-11-21
> > > > > > > **Rebuttal to Reviewer ixBu (Part 6)**
> > > > > > >
> > > > > > > We thank the reviewer for the detailed follow-up comments and for engaging so carefully with our work. Below we provide our responses.
> > > > > > > > **On Data Coverage & Overclaiming**
> > > > > > >
> > > > > > > We believe part of the disagreement comes from conflating data and method. **We agree with the reviewer that our current training set cannot cover all forms of OCR uncertainty, and we do not claim that it does so.** Our contribution is a training procedure rather than a specific degradation dataset: the cold-start + GRPO recipe is model- and domain-agnostic, and in this paper we instantiate it on degradation-induced uncertainty as a concrete, challenging testbed.
> > > > > > >
> > > > > > > For the reviewer’s “high-density formula” example: in any formula-heavy corpus where a strong VLM fails, our current pipeline would simply pseudo-label those misrecognized formula spans as UNC during cold start (via GT mismatch), and GRPO would then refine the model to consistently enclose such error spans with UNC tags.
> > > > > > >
> > > > > > > **Our claim is not that we already solve every uncertainty scenario, but that the same training recipe can, in principle, be applied to additional regimes (e.g., formulas, handwriting, ASR) once suitable data for those regimes is available.**
> > > > > > >
> > > > > > > > **On evaluation cost**
> > > > > > >
> > > > > > > We agree that inference-only evaluation on a new benchmark is computationally cheap and our earlier remark was not meant to suggest that GPU time is expensive. However, in your previous comment you explicitly suggested that “the authors could retrieve subsets with occlusion and other issues from these benchmarks.” We believe that the main cost lies in identifying such subsets and obtaining ground-truth annotations for uncertain regions. This cost comes from curation and annotation, not from the forward passes themselves.
> > > > > > >
> > > > > > > Moreover, our current model is a degradation-specialized proof-of-concept rather than a broadly trained, general-purpose OCR system. Simply running it on benchmarks such as OmniDocBench, without the above subset construction, would primarily measure plain OCR accuracy in a distribution it was not trained for, and would not provide a fair or informative assessment of the proposed uncertainty-tagging framework.
> > > > > > >
> > > > > > > *We have updated the conclusion section in pdf to explicitly state these points and to avoid potential misunderstandings.*

---

> > > > > > > > ### Comment · Reviewer_ixBu · 2025-11-22
> > > > > > > >
> > > > > > > > I appreciate the authors for clarifying the actual scope of the method's effectiveness. I feel that this work has the potential to receive a higher score after modifying the presentation to clearly define these boundaries. The title and motivation section should focus on the degraded doc OCR currently solved by **blur-OCR**, for instance, by **explicitly highlighting that this work solves degraded document OCR, or by framing it as an exploration of uncertainty OCR using degraded document OCR as a case study.** Regarding the results, I feel this method might partially generalize to other uncertainty scenarios, but neither the dataset preparation nor the evaluation covers these. Therefore, the work needs to move any uncertainty scopes that exceed the current experiments into the **Future Work** and **Discussion** sections. Fortunately, since this is ICLR, the presentation can be updated directly; for other conferences, such revisions—**e.g., manually selecting subsets to demonstrate generalization capabilities or changing the scope**—would have to wait for the next round of peer review.
> > > > > > > >
> > > > > > > > A specific reminder: although you mentioned `we do not claim that it does so`, the initial impression given to the reader is that the paper claims to cover a broad scope. A typically rigorous paper would include a **Figure 1** to demonstrate the specific scope of the problem being addressed, contrasting it with what other current VLMs for OCR cannot achieve. Additionally, based on my experience with pre-training and **GRPO** post-training, the degraded OCR scenario should not compromise general capabilities. You might find it beneficial to look into RLVR works like **Tulu 3** [1] for future reference.
> > > > > > > >
> > > > > > > > [1] Lambert, N., Morrison, J., Pyatkin, V., Huang, S., Ivison, H., Brahman, F., ... & Hajishirzi, H. (2024). Tulu 3: Pushing frontiers in open language model post-training. arXiv preprint arXiv:2411.15124.

---

> > > > > > > > > ### Author Response · Authors · 2025-11-22
> > > > > > > > > **Rebuttal to Reviewer ixBu (Part 7)**
> > > > > > > > >
> > > > > > > > > We thank the reviewer for the thoughtful follow-up and for sharing helpful references. We very much appreciate this discussion.
> > > > > > > > >
> > > > > > > > > > **The title and motivation section should focus on the degraded doc OCR currently solved by blur-OCR, for instance, by explicitly highlighting that this work solves degraded document OCR, or by framing it as an exploration of uncertainty OCR using degraded document OCR as a case study. Therefore, the work needs to move any uncertainty scopes that exceed the current experiments into the Future Work and Discussion sections. The initial impression given to the reader is that the paper claims to cover a broad scope.**
> > > > > > > > >
> > > > > > > > > In fact, the current PDF already restricts its main scope to degraded document OCR.
> > > > > > > > > 1. In the abstract, the phrase “degraded documents” appears three times, and throughout the main body we only discuss degraded document OCR, without claiming to address other types of OCR uncertainty.
> > > > > > > > > 2. (L14) Our motivation explicitly states: “To improve the trustworthiness of OCR on degraded documents, we propose uncertainty-aware OCR.”
> > > > > > > > > 3. (L497) Our conclusion explicitly states: “These results demonstrate that making uncertainty explicit offers a practical path to more trustworthy degraded document understanding.”
> > > > > > > > > 4. The only place where we mention broader uncertainty scenarios is in the Future Work section.
> > > > > > > > >
> > > > > > > > > We therefore believe that the current version already aligns with the reviewer’s suggestion.
> > > > > > > > >
> > > > > > > > >
> > > > > > > > > > **Based on my experience with pre-training and GRPO post-training, the degraded OCR scenario should not compromise general capabilities. You might find it beneficial to look into RLVR works like Tulu 3 for future reference.**
> > > > > > > > >
> > > > > > > > > Thank you for the pointer to Tulu 3. A key idea emphasized in their paper is **data mixing**, which helps mitigate catastrophic forgetting by continually exposing the model to general data.
> > > > > > > > >
> > > > > > > > > Our current work is a proof of concept in the degraded document OCR setting, so in both the cold-start SFT and GRPO stages we did not mix in additional general OCR data (such as complex layouts, mathematical formulas, etc.). As a result, some degradation of the backbone’s general OCR capabilities is expected. Extending our training recipe with appropriate data mixing to obtain a more general-purpose OCR model is an important direction for future work.
> > > > > > > > >
> > > > > > > > > *We sincerely appreciate the constructive discussion and thoughtful suggestions. We are happy to clarify any remaining questions or concerns and would welcome further feedback.*

---

> > > > > > > > > > ### Comment · Reviewer_ixBu · 2025-11-22
> > > > > > > > > >
> > > > > > > > > > Already? Really? I notice there are still a few days left. I hope the author can think about ways to improve their work for better results and cherish the opportunity of this discussion.

---

> > > > > > > > > > > ### Author Response · Authors · 2025-11-27
> > > > > > > > > > > **Rebuttal to Reviewer ixBu (Part 8)**
> > > > > > > > > > >
> > > > > > > > > > > We thank the reviewer for the constructive suggestions. In order to mirror the discussion in the rebuttal and make the main message of the paper easier for readers to follow, we have further revised several key parts of the paper:
> > > > > > > > > > >
> > > > > > > > > > > 1. We clarified that cold-start pseudo-labels are meant to tag erroneous model outputs, but not all text from visually degraded regions.
> > > > > > > > > > > 2. In the benchmark section, we now explain why we do not explicitly annotate “truly unreadable” regions.
> > > > > > > > > > > 3. We added more references on OCR and VLM uncertainty to better position our approach.
> > > > > > > > > > > 4. We explicitly narrow the scope to degraded documents.
> > > > > > > > > > >
> > > > > > > > > > > In addition, we note that Figure 1 is already in line with the reviewer’s suggestion: it illustrates the scope of the problem, what current VLMs for OCR cannot achieve, and how our method addresses this gap. The left side shows a base model hallucinating content on a noisy image without any indication of uncertainty, while the right side (with GRPO) shows our UNC model producing a transcript but adding UNC tags over spans where it is uncertain.
> > > > > > > > > > >
> > > > > > > > > > > We hope these revisions address the reviewer’s concerns and help improve the paper, and we would be grateful for any further feedback.

---

> ### Comment · Reviewer_ixBu · 2025-11-28
>
> Thank you for your response. I basically have no concerns now, except for adding "visually degraded documents" to the title. Considering that there was a good OCR paper on this angle at this year's ACL (main), and since the method in this manuscript is considered workable, I will change my rating to 6 points once the openreview system is fixed.
> \
> \
> \
> \
> \
> \
> \
> \
> \
> \
> \
> \
> \
> \
> \
> \
> \
> \
> \
> \
> \
> \
> \
> \
> \
> \
> \
> \
> \
> \
> \
> \
> \
> \
> \
> \
> \
> \
> \
> \
> \
> \
> \
> \
> \
> \
> However, if the title does not change, I might downgrade it to 2 points.       \: 》

---

> > ### Author Response · Authors · 2025-11-28
> >
> > We are glad that our responses have addressed the reviewer’s concerns.
> >
> > Following your suggestion, we have updated the title of the paper to “Teaching VLMs to Admit Uncertainty in OCR on Visually Degraded Documents” so that it more accurately reflects the scope and setting of our work.

---

> > > ### Comment · Reviewer_ixBu · 2025-11-28
> > > **To AC, all concern resolved and final score is 8.**
> > >
> > > Thank you for the response. I feel that this work deserves a **strong accept**. I hope this work is well-prepared for community use; I believe the algorithms and datasets are very helpful and could potentially lead to a direction in our community like search-r1 [1] in deep-research.
> > >
> > > I hope communities members can follow up on this work to verify whether it can further cover various VLM OCR uncertainty issues and ultimately produce unified vlm RLVR methods  for OCR or even token-level hallucination and calibration.
> > >
> > >
> > > To the AC, my final score is 8.  ： ）
> > >
> > >
> > > [1] Jin B, Zeng H, Yue Z, et al. Search-r1: Training llms to reason and leverage search engines with reinforcement learning[J]. arXiv preprint arXiv:2503.09516, 2025.

---

### Official Review · Reviewer_8tLK · 2025-10-30

**Soundness:** 2
**Presentation:** 3
**Contribution:** 3
**Rating:** 6
**Confidence:** 3

**Summary:**

The paper presents an uncertainty-aware fine-tuning framework for OCR-capable large language models (LLMs).
Instead of producing overconfident transcriptions for visually degraded documents, the model is trained to explicitly mark uncertain spans with specific tags.
The approach combines two stages: (1) a cold-start supervised fine-tuning (SFT) phase using pseudo uncertainty labels automatically derived from model errors, and (2) Group Relative Policy Optimization (GRPO), a reinforcement learning algorithm that jointly optimizes transcription accuracy and uncertainty tagging quality using a reward function balancing edit distance and F-beta-based span precision–recall.
Experiments on the new Blur-OCR benchmark demonstrate that GRPO improves both transcription correctness and uncertainty calibration compared to the cold-start baseline, while avoiding degenerate behaviors such as excessive tagging.

**Strengths:**

1. The problem is clearly motivated — OCR hallucination is a realistic and underexplored setting for uncertainty estimation.
2. The proposed explicit uncertainty tagging paradigm is conceptually simple yet effective.
3. The GRPO objective is well designed, balancing transcription accuracy and tagging F1, and preventing pathological behaviors via the reward-damping term.
4. The evaluation setup is thorough: they report both accuracy and uncertainty metrics and analyze different training stages.
5. The paper is very clearly written and easy to follow, with transparent motivation and mathematical detail.

**Weaknesses:**

1. Experiments are limited to a single OCR model family. It remains unclear whether the proposed method generalizes to other setups (e.g., multi-modal vision-language models such as Donut or TrOCR).
2. Comparison baselines are relatively narrow — no direct comparison with alternative uncertainty modeling approaches such as entropy-based rejection or calibration.
3. There is limited qualitative analysis of false-positive tags (over-tagging). Some visual examples or error breakdowns could strengthen interpretability claims.

**Questions:**

1. Did the authors try training with other values of λ — how sensitive are results to this hyperparameter?
2. How would the approach behave in non-OCR settings, such as code transcription or speech recognition?
3. Did you observe any tendency for the model to under-tag uncertainty after GRPO fine-tuning?

---

> ### Author Response · Authors · 2025-11-19
> **Rebuttal to Reviewer 8tLK**
>
> We thank Reviewer 8tLK for thoughtful review. Below is our rebuttal.
>
> > **W1. Experiments are limited to a single OCR model family.**
>
> We appreciate the reviewer’s suggestion. In the revised version, we add experiments with an additional backbone, InternVL2.5 (see Exp4). On this model, our uncertainty-aware training consistently improves UNC-tag F1 while preserving accuracy, demonstrating that our method can be applied to different models.
>
> > **W2. no direct comparison with alternative uncertainty modeling approaches such as entropy-based rejection or calibration.**
>
> In our initial submission (Exp. 4), we already included two standard uncertainty-quantification baselines: the first is based on token-level Shannon entropy, and the second is analogous to a “multi-VLM agreement” approach, reported as “Ensembles” in Exp. 4, where we run five general LLMs and estimate uncertainty from the disagreement among their outputs.
>
> On Blur-OCR, both baselines are clearly weaker than our approach (e.g., entropy: 0.41, ensemble: 0.49, ours: 0.69).
>
> > **W3 & Q3. Limited qualitative analysis of false-positive tags (over-tagging) and Did you observe any tendency for the model to under-tag uncertainty after GRPO fine-tuning?**
>
> In our experiments, we explicitly report UNC-tag precision and recall: precision reflects over-tagging (false positives), while recall reflects under-tagging (missed uncertain regions). Across all settings, GRPO improves or maintains both precision and recall compared to the cold-start model, so we do not observe a tendency to under-tag after fine-tuning. We also already provide qualitative visual examples in Appendix F.
>
> > **Q1. Did the authors try training with other values of λ**
>
> Yes. In our ablation study (Exp3), we trained models with a range of λ values, which empirically supports our theoretical requirement that λ < 1. We observe that λ mainly affects the relative convergence speed of accuracy and UNC-tag F1. Based on these results, we recommend choosing λ in the 0.5–0.9 range.
>
> >**Q2. How would the approach behave in non-OCR settings, such as code transcription or speech recognition?**
>
> Our experiments are instantiated in OCR. However, the underlying problem we study is more general. Our goal is to train LLMs to explicitly express uncertainty when their inputs are lossy or compressed representations of the original evidence, instead of hallucinating over-confident answers from incomplete information.
>
> This situation arises in many other settings. For example, in speech recognition from noisy or low-quality audio, an ASR-LLM system could mark acoustically ambiguous regions with UNC tags. Recent work such as DeepSeek-OCR compresses long visual contexts into visual tokens; these tokens can be viewed as a low-fidelity sketch of the original document, analogous in spirit to our degraded images, and our approach suggests a way for LLMs to produce more trustworthy outputs from such compressed representations by explicitly highlighting uncertain spans. More broadly, in QA systems that are required to produce an answer even under partial evidence, one could train the model to wrap low-confidence parts of its response with UNC tags rather than responding in an uncalibrated, fully confident manner.
>
> We view our method as a step toward uncertainty-aware generation from lossy inputs, and we hope it will encourage the community to explore these extensions in future work.

---

### Official Review · Reviewer_MjWd · 2025-11-03

**Soundness:** 3
**Presentation:** 2
**Contribution:** 3
**Rating:** 8
**Confidence:** 4

**Summary:**

This work tackles the problem of hallucinations for OCR for vision-language models. The models hallucinate when they are provided with blurry documents. This work proposed to use Reinforcement learning - GRPO to tackle this problem where the model is made to answer uncertainty tags along with transcription. They construct a multi-objective reward that balances accuracy with uncertainty and also mitigates reward hacking. They also provide a benchmark to measure uncertainty aware OCR performance on degraded documents and show that their method outperforms baselines.

**Strengths:**

- Uncertainty estimation has been a long studies problem but this work studies it in the context of OCR which is an important problem and the technique of using RL to is interesting.
- The work discusses the importance of cold-start SFT, describes in detail their reward formulation, character vs word level tagging, different hyperparameters and provide comprehensive experiments.

**Weaknesses:**

- The benchmark that they introduce has synthetic degradations. It is unclear how much the results transfer to actual degradations found in practice.
-  The paper has nice set of experiments but lacks some intuitions examples as detailed below.

**Questions:**

- The authors discuss tag validity and alignment around lines 113. I don’t fully understand how the GT segments which are not inside y^hat work. It would be nice to clarify with an example.
- In lines 370, the authors discuss the tradeoff of using character level vs word level granularity for tags. Could the authors provide an example along with the text? This would make it easier to follow. What do the authors mean by character level rewards?
- Can the authors compare their method to some baselines used in the uncertainty literature like softmax probabilities, entropy etc?

---

> ### Author Response · Authors · 2025-11-19
> **Rebuttal to Reviewer MjWd**
>
> We thank Reviewer MjWd for their positive and encouraging evaluation. Below is our rebuttal.
>
> > **W1. The benchmark that they introduce has synthetic degradations. It is unclear how much the results transfer to actual degradations.**
>
> We fully understand the reviewer’s concern about the gap between synthetic and real-world degradations. Our core objective is to teach an LLM to explicitly express uncertainty whenever the underlying visual evidence is missing or severely corrupted. As long as some regions are genuinely visually damaged, the task of “admitting uncertainty instead of hallucinating” remains meaningful, regardless of the exact degradation type. Our model is encouraged to associate ambiguous or low-fidelity regions with UNC tags, and this mechanism is largely agnostic to the exact type of degradation.
>
> Practically, we are not aware of any large-scale corpus of real degraded unreadable documents with high-quality ground-truth transcripts. This motivated us to construct Blur-OCR using synthetic degradations. The degradation pipeline is intentionally highly stochastic: it includes 15 degradation operations with randomized parameters and random ordering, so that a wide variety of real-world artifacts are covered as special cases. Prior work (e.g., PreP-OCR) has shown that such synthetic degradations can transfer effectively to real degraded documents. In addition, we qualitatively evaluated our UNC model on a small set of real degraded documents and observed very similar UNC behavior to that on Blur-OCR. We agree that a larger real-world benchmark would further strengthen our claims and plan to extend Blur-OCR with real degraded documents in future work, once an appropriate set of documents with ground-truth has been identified.
>
> > **Q1 & Q2. Don’t fully understand how the GT segments which are not inside y^hat work. Could the authors provide an example along with the text? What do the authors mean by character level rewards?**
>
> Our initial submission already includes some examples of GT–y^hat alignment at both character and word level in Appendix C, and we also provide the Python code for alignment and inserting UNC tags in the supplementary material. In our first example in Appendix C:
>
> GT: The quick brown fox jumps
>
> y^hat: Te quick brownnn fox jumaps
>
> Here the character “h” in “The” is missing, so we place the word closest to that “h” character inside the UNC tag. Thus, the character-level and word-level alignments are respectively:
>
> Char: {C}Te{/C} quick brown{C}nn{/C} fox jum{C}a{/C}ps
>
> Word: {C}Te{/C} quick {C}brownnn{/C} fox {C}jumaps{/C}
>
> All of our metrics and rewards can be computed either at the character level or at the word level. A character-level reward simply treats each character as the atomic unit when computing the reward (see Section 4 & 5.2), while the word-level reward treats each whitespace-separated token as the unit.
>
> > **Q3. Can the authors compare their method to some baselines used in the uncertainty literature like softmax probabilities, entropy etc?**
>
> In our initial submission (Exp. 4), we have already included two standard uncertainty-quantification baselines: the first is based on token-level Shannon entropy, and the second is analogous to a “multi-VLM agreement” approach, reported as “Ensembles” in Exp. 4, where we run five general LLMs and estimate uncertainty from the disagreement among their outputs.
>
> On Blur-OCR, both baselines are clearly weaker than our approach (e.g., entropy: 0.41, ensemble: 0.49, ours: 0.69).

---

### Official Review · Reviewer_29Ew · 2025-11-05

**Soundness:** 2
**Presentation:** 3
**Contribution:** 2
**Rating:** 6
**Confidence:** 4

**Summary:**

This paper focuses on a task: when performing OCR, a VLM should mark uncertain and hard-to-recognize spans by surrounding them with a custom UNC tag. To enable this, the authors build a training set of about 100K samples and a 2K-sample benchmark. For training, they use two phases — SFT for warm-start, followed by RL.

**Strengths:**

1. The writing is clear, and the presentation of different settings is easy to follow.

2. Introducing uncertainty-aware generation in OCR to mark unclear spans has practical value.

3. The experiments include extensive ablation study, which helps clarify the effectiveness of the method, and I appreciate that.

**Weaknesses:**

1. Minor: Although admitting uncertainty in OCR has some practical value, on the other hand, the broader significance is also limited since the work focuses on a specific application setting.

2. The backbone choice is quite limited.

3. The benchmark is constructed by the authors, while appreciated, i also want to know how well the method (and models) generalizes to more OOD scenarios.

4. Minor: I think the paper should use “VLM” instead of “LLM.”

5. While I appreciate the detailed metric section and the following ablation studies, it feels somewhat over-emphasized. The data component should likely deserve more effort than defining the metrics.

**Questions:**

is it possible to do semi-supervised learning on this task since uncertainty can also be generated by LLM itself (although it may not be so well-calibrated)?

---

> ### Author Response · Authors · 2025-11-19
> **Rebuttal to Reviewer 29Ew**
>
> We thank Reviewer 29Ew for thoughtful review. Below is our rebuttal.
>
> > **W1. Broader significance is limited since the work focuses on a specific application setting.**
>
> We agree that our experiments are instantiated in OCR. However, the underlying problem we study is more general. Our goal is to train LLMs to explicitly express uncertainty when their inputs are lossy or compressed representations of the original evidence, instead of hallucinating over-confident answers from incomplete information.
>
> This situation arises in many other settings. For example, in speech recognition from noisy or low-quality audio, an ASR-LLM system could mark acoustically ambiguous regions with UNC tags. Recent work, such as DeepSeek-OCR, compresses long visual contexts into visual tokens. These tokens can be viewed as a low-fidelity sketch of the original document, analogous to our degraded images, and our approach suggests a way for LLMs to produce more trustworthy outputs from such compressed representations by explicitly highlighting uncertain spans. More broadly, in QA systems that are required to produce an answer even under partial evidence, one could train the model to wrap low-confidence parts of its response with UNC tags rather than responding in an uncalibrated, fully-confident manner.
>
> We view our method as a step toward uncertainty-aware generation from lossy inputs, and we hope it will encourage the community to explore these extensions in future work.
>
> > **W2. Backbone choice is quite limited**
>
> To address this concern, we added further experiments with an additional backbone, InternVL2.5, in Exp4. We show that our uncertainty-aware training consistently improves UNC-tag F1 while also preserving transcription accuracy.
>
> > **W3. How well the method (and models) generalizes to more OOD scenarios.**
>
> We are not aware of any public large-scale corpus of real degraded documents with high-quality ground-truth transcripts. This motivated us to construct Blur-OCR using synthetic degradations. Prior work (e.g., PreP-OCR) has shown that such synthetic degradations can transfer effectively to OOD scenarios. In addition, we qualitatively evaluated our UNC model on a small set of real degraded documents and observed very similar UNC behavior to that on Blur-OCR.
>
> > **W4. I think the paper should use “VLM” instead of “LLM.”**
>
> Thank you for the suggestion. We will consider using the term “VLM.”
>
> > **W5. The data component should likely deserve more effort than defining the metrics.**
>
> The metrics are indeed described in detail because they are tightly coupled with our GRPO reward and are central to evaluating both transcription accuracy and span-level UNC behavior. But we agree that the data component also deserves more space. In the revised version, we add further description of Blur-OCR (degradation types and parameter ranges) in Section 6 and the appendix, so that both the benchmark construction and the evaluation protocol are more clearly documented.
>
> > **Q1. Is it possible to do semi-supervised learning on this task since uncertainty can also be generated by LLM itself (although it may not be so well-calibrated)?**
>
> Yes, this could be feasible and worth investigating. We could potentially use the model's intrinsic uncertainty (e.g., entropy or ensemble disagreement) to generate pseudo-labels on unlabeled real-world data. However, given that raw model confidence is often miscalibrated, as the reviewer rightly pointed out, we have prioritized the supervised setting in this work to establish a reliable upper bound and validate the training strategy before tackling the noise that is inherent in semi-supervised learning.

---

### Author Response · Authors · 2025-12-03
**Rebuttal Summary for Submission 1052**

Dear Reviewers & AC,

We would like to thank all reviewers for their constructive feedback and the time devoted to our submission. We are also grateful to the AC for coordinating this review during a particularly challenging cycle.

Across the reviews, there is a clear consensus that the task we study is important and well-motivated.

For Reviewers 29Ew, MjWd, and 8tLK, we responded point-by-point to their listed weaknesses and questions. Although they did not further reply, we believe our clarifications make our setting and experiments easier to understand. With Reviewer ixBu, we had multiple rounds of detailed discussion, mainly around benchmark choice and the cold-start procedure. Based on this, we added an additional backbone and new baselines, tightened and clarified several parts of the paper, and made the main messages of some sections easier for readers to follow. The reviewer ultimately stated that their concerns were resolved and raised the final score to 8.

Some reviewers also asked whether our training paradigm can generalize beyond degraded document OCR. Our current experiments are focus on visually degraded document OCR, but we think that the same uncertainty-aware training approach could be applied to other lossy or ambiguous input settings (e.g., handwritten documents, noisy speech transcripts, or compressed visual tokens). We believe our work opens up many promising directions for subsequent research.

Best regards,
1052 Authors

---

### Meta-Review · Area_Chair_aY7D · 2026-01-08

**Summary:**

Reviewers agree the paper tackles an important, practical failure mode—VLM-based OCR hallucinating on visually degraded documents—by training models to explicitly tag unreliable spans, and the empirical results (including ablations, reward design against hacking, and a new Blur-OCR benchmark) support that the method improves uncertainty tagging while largely preserving transcription accuracy.

**Reviewer Concerns:**

The rebuttal largely resolves the main concerns: added an additional backbone (addressing limited model scope), clarified cold-start intent and uncertainty definition, included/clarified entropy and ensemble baselines, added qualitative examples and precision/recall evidence for over/under-tagging, and—crucially—tightened terminology/scope (VLM vs LLM; degraded-doc OCR focus), which directly addressed ixBu’s strongest objections; the remaining open point is limited real-world degradation coverage/benchmark breadth, but this is acknowledged and reasonably positioned as future work given annotation/curation constraints.

**Reviewer Scores:**

Three reviewers remains positive and Reviewer ixBu explicitly updates 4→8 after title/scope/terminology fixes and discussion.

---

### Decision · Program_Chairs · 2026-01-26

Accept (Poster)